# Glycolytic flux sustains human Th1 identity and effector function via STAT1 glycosylation

Ariful Haque Abir[1,2,3,4] , Julia Benz[5], Benjamin Frey[6], Heiko Bruns[7] , Gerhard Krönke[2,3,4], Udo S Gaipl[6], Kilian Schober[5,8], Dimitrios Mougiakakos[9], Dirk Mielenz[1,8,10]

T helper (Th) cell lineages are linked to metabolism, but precise mechanisms in human Th1 cells remain unclear. We addressed this question by in vitro stimulation and CRISPR/Cas9-mediated gene editing. Metabolic profiling revealed enhanced glycolytic activity in Th1 versus non-polarized cells, evidenced by increased extracellular acidification rate, ATP production via glycolysis, lactate secretion, NADH abundance, and elevated glycolysis-dependent anabolic activity. Inhibition of glycolysis reduced IFNγ production and STAT1 phosphorylation independent of JAK1/2 or SHP2 activity and STAT1 abundance, implicating glycolysis directly in sustaining STAT1-mediated Th1 functionality. O-glycosylation of STAT1 via O-glycosyltransferase was pivotal in modulating STAT1 activity. Pharmaceutical O-glycosyltransferase inhibition prevented Th1 differentiation as well as STAT1 O-glycosylation. CRISPR/Cas9 mediated mutation of the O-glycosylation sites at Ser499 and Thr510 diminished STAT1 Ser727 phosphorylation and IFNγ synthesis. Together, this study highlights glycolysis as key regulator of human Th1 cell identity and effector function, with STAT1 O-glycosylation selectively maintaining Th1 effector capacity. This mechanism could be explored to safeguard Th1 cells.

## Introduction

Naïve CD4[+] T cells must receive signals from pMHC via CD3 and costimulation via CD28 for full activation (Kumar et al, 2018). The latter triggers anabolic phosphoinositol-3-kinase/Akt/mammalian target of rapamycin complex (PI3K/Akt/mTORC) 1 signaling, which is required for T cell growth and effector functions (Ho et al,

2009; Johnson et al, 2018). Seminal observations revealed that there are different Th effector cell types based on cytokine profiles (Mosmann et al, 1986). Th1 cells produce IFNγ, IL-12, and TNFα whereas Th2 cells generate IL-4, IL-5, IL-13; Th9 cells express IL-9 and Th17 cells induce IL17 (Golubovskaya & Wu, 2016). IFNγ producing Th1 cells protect from intracellular bacteria, parasites, viruses, and cancer. The downside is their involvement in autoimmunity (Annunziato & Romagnani, 2009; McNab et al, 2015). Thus, mechanisms controlling Th1 homeostasis and effector capacity, that is, IFNγ production, are of considerable interest. T-bet (TBX21) is the master transcription factor for Th1 cells and controls IFNγ production (Szabo et al, 2000, 2002). It reinforces its own expression, thereby consolidating the Th1 phenotype (Lighvani et al, 2001; Afkarian et al, 2002). IL-12 has a pivotal upstream role in Th1 differentiation (Trinchieri et al, 1992; Hsieh et al, 1993) by inducing T-bet expression and Th1 polarization independent of IFNγ signaling. Hence, the autoregulation of T-bet becomes superfluous in the presence of IL-12 or IFNγ (Mullen et al, 2001). T-bet maintains Th1 identity by repressing the Th2 defining transcription factor GATA3 and vice versa (Usui et al, 2006). However, hybrid Th1/Th2 cells have been described in mice (Peine et al, 2013).

During Th1 differentiation, signal transducer and activator of transcription (STAT) 1 and Janus kinase (JAK) 1/2 take center stage (Wilks, 1989; Wilks et al, 1991; Shuai et al, 1993). STAT1 is essential for Th1 cell differentiation and the production of IFNγ. Phosphorylation of STAT1 via IL-12 mediated JAK2 activation at Tyr701 is the first step in the activation process (Barnholt et al, 2009; Seif et al, 2017); it enables the nuclear translocation of STAT1. Subsequent phosphorylation of STAT1 at Ser727 via autocrine IFNγ receptor signaling is essential for full transcriptional activation. JAK-mediated phosphorylation of STAT is a prerequisite to induce IFN-associated gene expression (Müller et al, 1993). In addition to phosphorylation at Tyr701 and Ser727, activation of STAT1 requires

[1]Division of Molecular Immunology, Department of Internal Medicine 3, Universitätsklinikum Erlangen und Friedrich-Alexander-Universität Erlangen-Nürnberg, Nikolaus-Fiebiger-Center, Erlangen, Germany [2]Medizinische Klinik mit Schwerpunkt Rheumatologie und Klinische Immunologie, Charité—Universitätsmedizin Berlin, Berlin, Germany [3]Deutsches Rheuma-Forschungszentrum Berlin, Berlin, Germany [4]Fraunhofer Institute for Translational Medicine and Pharmacology ITMP, Immunology and Allergology IA, Berlin, Germany [5]Mikrobiologisches Institut – Klinische Mikrobiologie, Immunologie und Hygiene, Universitätsklinikum Erlangen und Friedrich-Alexander-Universität (FAU) Erlangen-Nürnberg, Erlangen, Germany [6]Translational Radiobiology, Department of Radiation Oncology, Universitätsklinikum Erlangen, Friedrich-Alexander-University Erlangen-Nürnberg, Erlangen, Germany [7]Medizinische Klinik 5—Hämatologie und Internistische Onkologie, Lehrstuhl für Hämatologie/Internistische Onkologie, Universitätsklinikum Erlangen, Erlangen, Germany [8]FAU Profile Center Immunomedicine, FAU Erlangen-Nürnberg, Erlangen, Germany [9]Department for Hematology, Oncology, and Cell Therapy, Otto-von-Guericke University Magdeburg, Magdeburg, Germany [10]Department of Translational Immunology, Universitätsklinikum Erlangen und Friedrich-Alexander-Universität (FAU) Erlangen-Nürnberg, Nikolaus-Fiebiger-Center, Erlangen, Germany

Correspondence: Dirk.Mielenz@fau.de

methylation of arginyl residues at the amino terminus which extends the half-life of STAT1 tyrosine phosphorylation (Subramaniam et al, 2001). Notably, Ser727 phosphorylation is independent of Tyr701, although both events are part of the IFNγ pathway (Zhu et al, 1997). Two splicing isoforms of STAT1 have been identified: STAT1 α (91 kD) and β (84 kD). Only STAT1α contains a full-length transcriptional activation domain with phosphorylation sites at Tyr701 and Ser727. In contrast, the STAT1β isoform has a truncated transcriptional activation domain and bears a single phosphorylation site at Tyr701 (Schindler et al, 1992; Müller et al, 1993).

CD4$^+$ T cells mainly depend on OxPhos during their development (Kouidhi et al, 2017). However, activation via the PI3K/Akt/mTORC1 axis shifts the cells towards glycolysis (Waickman & Powell, 2012; Kouidhi et al, 2017). Of note, STAT1 transcriptionally activates glycolytic genes such as ENO1 and PDK3, further enhancing glycolysis, in human mesenchymal stem cells (MSCs) (Jitschin et al, 2019). Glycolysis provides, on the one hand, rapid energy in the form of ATP and, on the other hand, supplies anaplerotic intermediates for redox balance and the hexosamine biosynthesis pathway. Major posttranslational protein modifications supported by the hexosamine pathway are protein N- and O-glycosylation (Chiaradonna et al, 2018). Two enzymes, O-glycosyltransferase (OGT) and O-Glycanase (OGA), add or remove O-glycosylation (Abramowitz & Hanover, 2018). O-glycosylation is increased during human T cell activation, suggesting dominant OGT activity during this process (Lund et al, 2016).

Most studies focusing on Th1 cell metabolism are based on murine models. Given the importance of STAT1 in human Th1 cell function, and considering the metabolic implications of STAT1 signaling in MSC, we explored the intercalation of metabolism and STAT1 in human Th1 cells. We show that glycolytic flux drives human Th1 cell differentiation via STAT1 activity. OGT-mediated glycosylation of STAT1 supports STAT1 phosphorylation at Ser727 in activated T cells, thereby stabilizing intracellular IFNγ production. Thus, glycolytic flux maintains OGT activity and STAT1 O-glycosylation to sustain human Th1 cell effector. This finding could help understand Th1 vulnerability in acute chronic viral infections or cancer, or assist Th1 targeted therapies in autoimmunity.

# Results

## STAT1 activity and IFNγ production distinguish Th1 from activated human T cells

To understand how metabolism and STAT1 activity are connected in human Th1 cells, we used in vitro differentiated Th1 cells (see Fig S1A–C for the complete gating strategy). Polarized Th1 cells were compared with non-polarized, only anti-CD3/CD28 activated T cells (Act.T) and Th1 cells were identified by IFNγ staining (Fig 1A and B). Th1 cultures typically contained ~3.5-fold more IFNγ$^+$ cells than Act.T cells, whereas IL-4 or IL-17 were absent and T-bet was more abundant in the IFNγ$^+$ cells (Fig 1C–G). Phospho-flow analysis of

the IFNγ$^+$ population revealed the presence of high levels of STAT1 pTyr701 and pSer727 compared with Act.T cells (Fig 1H–J). To corroborate the function of STAT1 in our culture system, we induced a null mutation in the stat1 gene in Th1 cultures by CRISPR/Cas9-mediated gene editing. Edited Th1 cultures contained lower frequencies of IFNγ positive cells and their IFNγ abundance was also lower (Fig 1K). Together, these findings reinforce the involvement of STAT1 in human Th1 differentiation and effector function.

## Glycolytic flux contributes to Th1 metabolism upon differentiation

Murine Th1 cells shift towards aerobic glycolysis in vitro (Palmer et al, 2015; Kouidhi et al, 2017). To characterize metabolic changes during human Th1 cell differentiation, we performed extracellular flux (Seahorse) assays of naïve, Act.T, and Th1 cell cultures. To be able to align these and the following experiments, we used the same conditions as for the IFNγ staining, that is, 4 h of PMA/ionomycin stimulation. The extracellular acidification rate (ECAR) was up-regulated in both Act.T and Th1 cells upon glucose addition. ECAR was slightly more elevated in Th1 culture conditions (Fig 2A and B; corresponding oxygen consumption rate data in Fig S2A and B). This mildly elevated glycolytic capacity and glycolytic reserve of Th1 cells indicated a higher rate of converting glucose into lactate (Fig 2C and D). ATP rate assays confirmed that Act.T, but more so Th1 cells, produce their maximum ATP through glycolysis rather than by mitochondrial metabolism (Fig 2E; corresponding oxygen consumption rate data in Fig S2A and B). To support these data on the single cell level, flow cytometry-based single cell metabolic analysis was performed via SCENITH (Single Cell ENergetic metabolism by profiling Translation inHibition) in the same culture conditions (Argüello et al, 2020). SCENITH depicts the dependence of protein synthesis rate on the available ATP pool. The source of ATP is thereby quantified by measuring puromycin incorporation as a surrogate for protein synthesis, based on the assumption that protein translation is the most energy-demanding cellular process. Interestingly, in the absence of inhibitors, Th1 cells identified by IFNγ exhibited more puromycin incorporation compared with Act.T (Fig 2F). This held true in separate cultures (Fig 2F(i)) and within the same culture (Fig 2F(ii)). However, puromycin incorporation depended on glycolysis in both Th1 and Act.T cells (Fig 2G), much more so than on cellular respiration, which is blocked when oligomycin was used (Fig 2H). Both Th1 and T.act cells depended less on fatty acid and amino acid oxidation than N.CD4$^+$ cells (Fig 2H). Consistent with the Seahorse findings, these data support the hypothesis that anabolism of human Th1 cells, but also Act T cells, is particularly fueled by glycolysis. To substantiate these findings, we performed a surrogate glucose uptake assay. After PMA/ionomycin restimulation, the cells were incubated in glucose-free medium with 6-NBDG (6-(N-(7-Nitrobenz-2-oxa-1,3-diazol-4-yl)amino)-6-Deoxyglucose), a fluorescent glucose analog, for 30 min. Th1 cells showed only a slightly higher median fluorescence intensity (MFI) for 6-NBDG (Fig 2I). However, glucose and lactate abundance in the culture medium on day 5 of the culture period revealed that Th1 cultures contained much less glucose and more lactate, evidencing a strong glycolytic activity (Fig 2J). These

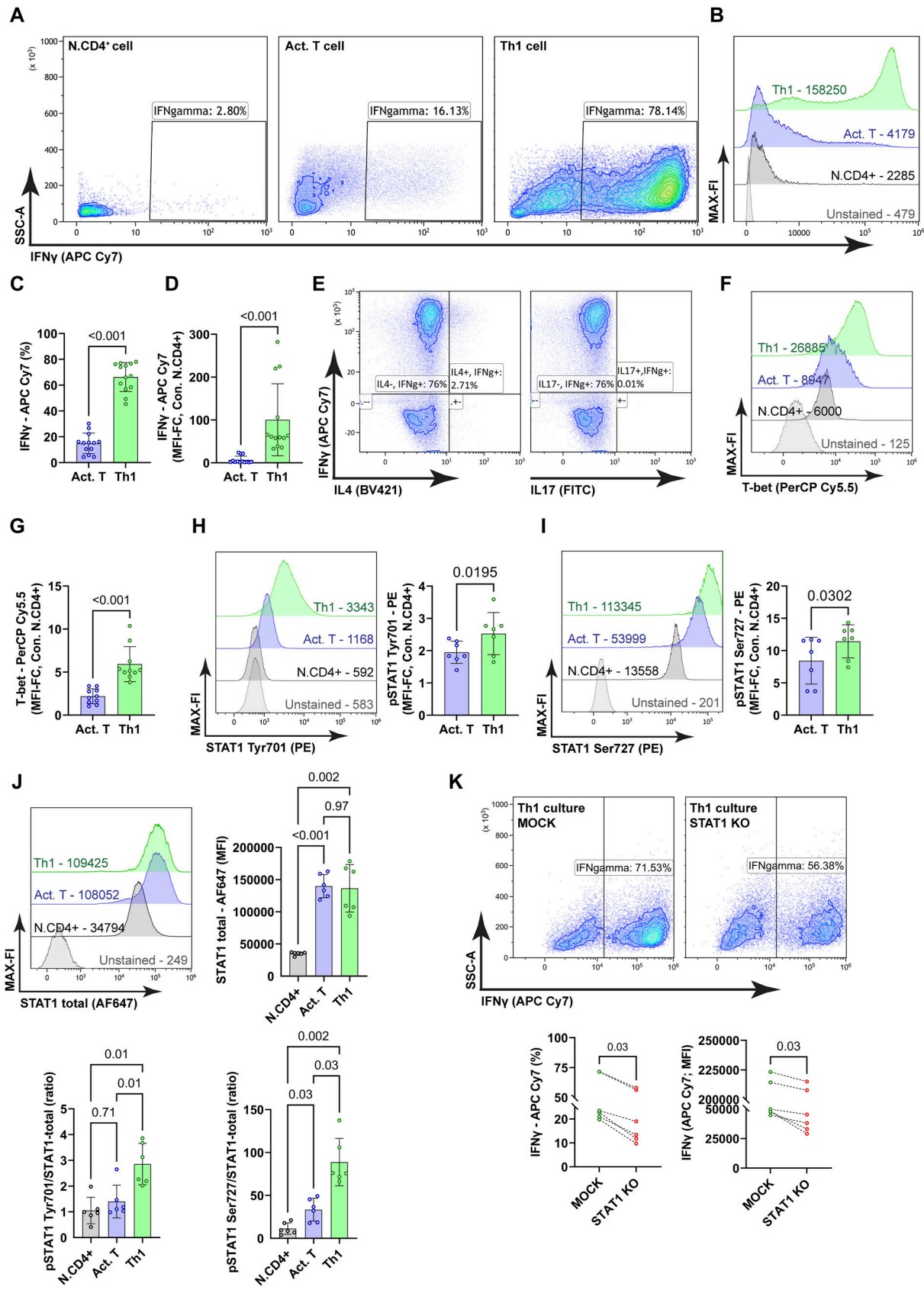

data suggested that the small differences in surrogate glucose uptake are disproportional in relation to the metabolic consequences, which might be explained by receptor-independent uptake of glucose (Hamilton et al, 2021). To reconcile these findings, we monitored glucose-driven NADH generation in real time by flow cytometry after adding glucose to starved cells (Abir et al, 2024) (Fig 2K), demonstrating that acute glycolytic flux is higher in Th1 than Act.T cells. Together, human Th1 cells exhibit a high biometabolic activity biased toward glycolysis, with excess glucose turnover, suggesting a relationship between human Th1 differentiation and glycolysis.

### Glycolytic flux dictates Th1 effector function via STAT1 phosphorylation

To uncover the relationship between Th1 differentiation, that is, IFNγ production, and glycolysis, Th1 cells were treated for 30 min with 2-DG and oligomycin, either separately or in combination, and IFNγ abundance was analyzed by flow cytometry. This analysis demonstrated that acute blockade of glycolytic flux with 2-DG notably reduced IFNγ production, without a discernible impact of oligomycin (Fig 3A). Furthermore, 2-DG diminished the frequency of IFNγ-producing cells (Fig 3B). These data indicate that glycolytic flux is essential to maintain both, Th1 identity as defined by the frequency of IFNγ-producing cells and effector capacity, defined as intracellular IFNγ amount. 2-DG acts upstream in the glycolytic pathway (Fig S3A). To corroborate our results, we used two additional inhibitors of glycolysis: Iodocetate (IAA), which inhibits glyceraldehyde-3-phosphate-dehydrogenase, and oxalate (OXA), which inhibits pyruvate kinase (Fig S3A). Notably, both IAA and OXA reduced the frequency of IFNγ+ cells (Fig S3B). IAA diminished STAT Tyr701 phosphorylation (Fig S3C) whereas IAA and OXA diminished STAT Ser727 phosphorylation only by trend (Fig S3D). However, both IAA and OXA effectively curtailed O-glycosylation (Fig S3E).

The phosphorylation of STAT1 is a critical step in the signaling cascade that is necessary for the production of IFNγ in Th1 cells (Shuai et al, 1993; Seif et al, 2017). Accordingly, we investigated the impact of 2-DG on STAT1 activation. There was a clear reduction in STAT1 phosphorylation at both Tyr701 and Ser727 in response to glycolytic blockade (Fig 3C and D). 2-DG and IAA reduced STAT Tyr701 whereas only 2-DG diminished STAT pS727 (Fig 3C and D). To ascertain that these alterations were not influenced by upstream signaling or altered total STAT1 abundance, we measured total STAT1 levels and JAK1/2 phosphorylation at the activating residues

Tyr1034/1035 and Tyr1007/1008, without significant changes (Fig 3E–G). These findings indicate that glycolysis plays a role in enhancing IFNγ production by modulating STAT1 phosphorylation downstream of JAK1/2 and independent of STAT1 abundance.

### 2-DG-induced STAT1 dephosphorylation partially depends on SHP2

Net phosphorylation is a balance between kinase and phosphatase activity. Src homology region 2 domain-containing phosphatase-2 (SHP2) can act as a dual-specific phosphatase in the case of STAT1, removing phosphorylation at both Tyr701 and Ser727, and 2-DG may intrinsically activate SHP2 (Wu et al, 2002; Chen et al, 2020). This could potentially contribute to the reduction of phospho-STAT1 in 2-DG-treated cells rather than the inhibition of glycolytic flux itself (Fig 4A). To test this possibility, Th1 cells were stimulated with PMA/ionomycin, with 2-DG, with two SHP2-specific inhibitors, TNO155 and RMC4550 (Pandey et al, 2019; LaMarche et al, 2020), or co-treated (Fig 4A). Unexpectedly, SHP2 inhibition alone did not elevate the phosphorylation of STAT1 at Tyr701 and Ser727, but it did so slightly after 2-DG treatment; however, this elevation of phosphorylation did not compensate for the inhibition by 2-DG (Fig 4B and C). Although unlikely, there might be no basal SHP2 activity under the used conditions because SHP2 is suppressed by PMA/Ionomycin through oxidation by ROS, or S-nitrosylation of the catalytic cysteine (Barrett et al, 2005; Pérez-Fernández et al, 2019) which might be overcome by 2-DG treatment. In addition, SHP2 inhibition did not reverse the reduction of IFNγ-producing cells mediated by treatment with 2-DG (Fig 4D and E). These findings indicate that besides SHP2 function, additional mechanisms downstream of glycolysis enforce STAT1 activity.

### Glycolysis controls Th1 differentiation via STAT1 O-glycosylation

To identify an additional mechanism by which glycolytic flux can influence STAT1 activity, we considered the hexosamine biosynthesis pathway, which contributes the intermediate GlcNac to the O-glycosylation pathway. First, we assessed the total O-glycosylation status by intracellular staining with the anti-O-glycosylation antibody RL2 (Jitschin et al, 2019). As described, T cell activation (Lund et al, 2016), but also Th1 differentiation increased global O-glycosylation compared with naive T cell counterparts (Fig 5A), insinuating that O-glycosylation is mainly induced by activation and not by Th1 differentiation. Moreover, the abundance

---

**Figure 1. Analysis of IFNγ production and STAT1 Phosphorylation upon Th1 differentiation.**
Naïve CD4+CD45RA+ T cells were activated by plate-bound anti-CD3 and anti-CD28 antibodies with IL-2 and anti-IL-4 antibodies (Act.T). Th1 differentiation was induced by additional IL-12. After 72 h, cells were restimulated with PMA and Ionomycin for 4 h prior analysis. Intracellular staining was performed, and cells were analyzed by flow cytometry. **(A)** Representative contour plots with IFNγ producing cells. **(B, C, D, E)** Representative histograms showing median fluorescence intensities (MFIs) (C) frequencies of IFNγ producing cells, mean ± SD, Wilcoxon matched-pairs test, n = 13 donors, (D) relative IFNγ abundance (Th1 versus Act.T), mean ± SD, Wilcoxon matched-pairs test, n = 13, (E). Th1 polarized cultures were intracellularly stained with anti-IL-4 conjugated with Brilliant Violet 421 (BV421) and anti-IL-17 conjugated with fluorescein isothiocyanate (FITC) and analyzed with flow cytometry. Representative contour plots are depicted. **(F)** Th1 polarized cultures were intracellularly stained with anti-T-bet antibody conjugated with Peridinin chlorophyll protein-Cyanine5.5 (PerCP Cy5.5) and analyzed by flow cytometry, a representative histogram with MFI values. **(G)** Relative T-bet abundance (Act.T and Th1 cells va. N.CD4+), mean ± SD, Wilcoxon matched-pairs test, n = 10. **(H, I)** STAT1 pTyr701 and pSer727 abundance, representative histograms and statistical analysis, mean ± SD, Wilcoxon matched-pairs test, n = 7. **(J)** STAT1 total abundance, representative histograms, pSTAT1/STAT1 total ratio ([pSTAT1/STAT1 total] × 100), and statistical analysis, mean ± SD, one-way ANOVA repetitive measures, n = 6. **(K)** Th1 cultured cells were electroporated with STAT1-targeting Cas9-gRNA ribonucleoproteins (RNPs) or MOCK electroporated, and cells were further cultured for 3 d with polarizing medium. **(J)** Representative contour plots, frequencies of IFNγ producing cells and IFNγ abundance, Wilcoxon matched-pairs test, mean ± SD, n = 6.

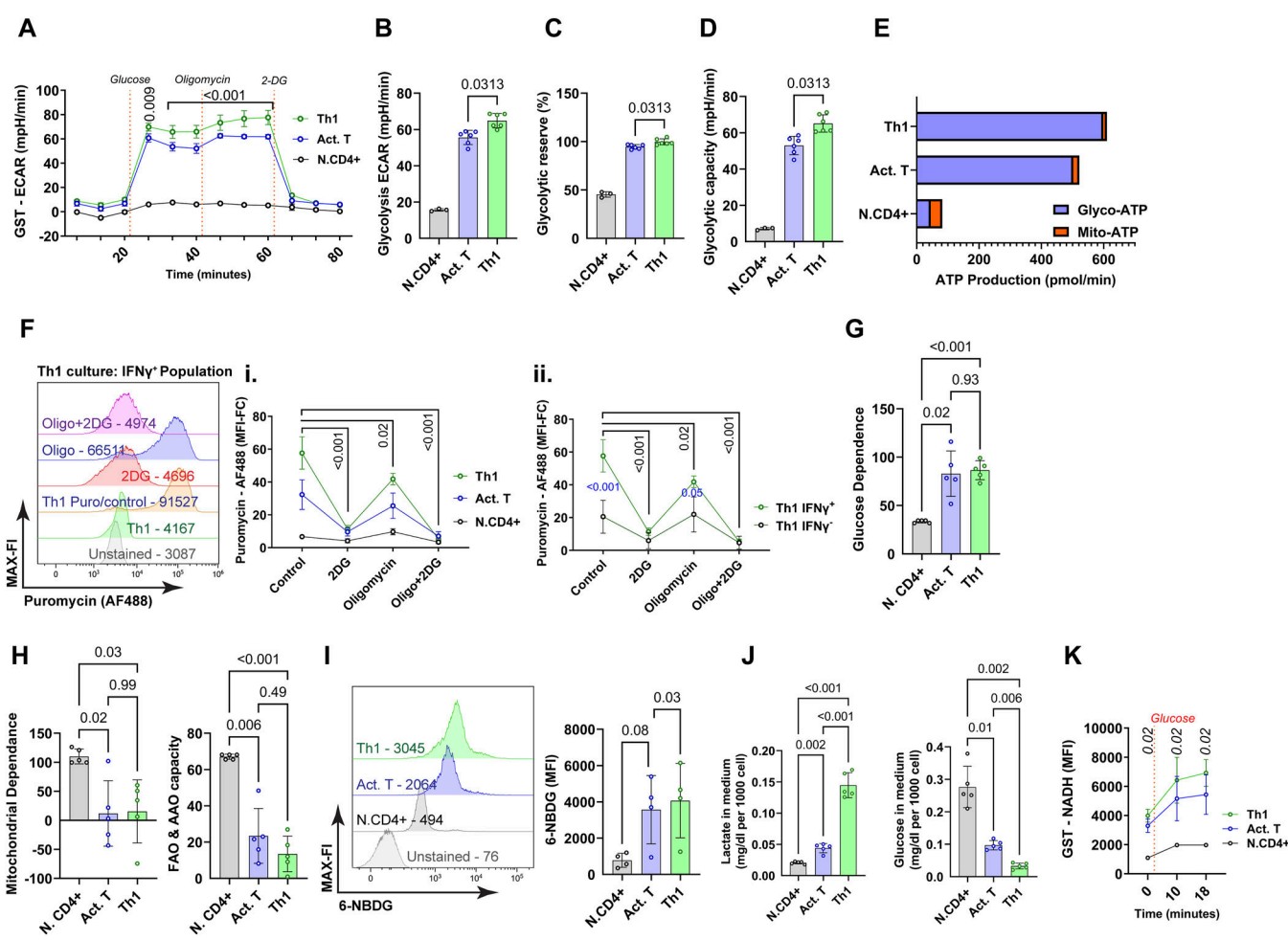

**Figure 2. Metabolic characterization of human Th1 cells.**
**(A)** N.CD4+, Act.T, and Th1 cells were harvested after PMA and ionomycin stimulation, and extracellular flux analysis was performed. The extracellular acidification rate (ECAR) was measured using a Seahorse XF96 device using the Seahorse glycolysis stress test (GST) by serial injection of glucose, oligomycin, and 2-deoxy-D-glucose (2-DG). Statistical analysis was performed with two-way ANOVA repetitive measures, the black bar with *P*-value indicates the significance of the difference between Act.T and Th1 cells between 30 to 60 min, mean ± SD, n = 6. **(B)** ECAR (after glucose and before oligomycin injection), Wilcoxon matched-pairs test, mean ± SD, n = 6. **(C, D, E)** Glycolytic reserve, Wilcoxon matched-pairs test, (D) Glycolytic capacity, Wilcoxon matched-pairs test, (E) Seahorse ATP rate assay. **(F(i), F(ii))** Representative histograms showing FI of anti-puromycin antibody conjugated with Alexa Fluor 488 (AF488) in different IFNγ+ populations in Th1 culture conditions; data are shown as mean ± SD two-way ANOVA repetitive measures, the black bar with *P*-value indicates the significance of the difference of Th1 cells between controls and different inhibitors, n = 5, (F(ii)) comparison of IFNγ+ and IFNγ- cells in Th1 cultures. **(G, H)** Glucose dependence, (H) mitochondrial dependence, and fatty acid oxidation (FAO) and amino acid oxidation (AAO) were calculated using the SCENITH data, one-way ANOVA repetitive measures (G, H), n = 5. **(I)** Glucose uptake was measured by flow cytometry using 6-NBDG (6(N-(7-Nitrobenz-2-oxa-1,3-diazol-4-yl)amino)-6-Deoxyglucose). MFI, value indicated in the representative histogram, statistics were analyzed with one-way ANOVA, repetitive measures, n = 4, mean ± SD. **(J, K)** Medium was collected on the final day of culture after PMA and Ionomycin stimulation, and glucose and lactate were quantified and normalized to cell numbers, one-way ANOVA repetitive measures, mean ± SD, n = 5. **(K)** NADH fluorescence was measured after the addition of glucose, two-way ANOVA repetitive measures, n = 3, mean ± SD.

of OGT was also augmented in both T.Act and Th1 cells (Fig 5B). However, this apparently does not exclude that O-glycosylation plays a role in Th1 identity, for instance, by targeting different substrates.

To clarify the proportional relation between Th1 differentiation, O-glycosylation and glycolysis, Th1 cells were treated with 2-DG, IAA, and OXA, and subjected to the same analysis. 2-DG clearly reduced total O-glycosylation in Th1 cells (Fig 5C), whereas IAA and OXA were less effective (Fig S3E). Pharmacological inhibition of OGT was then accomplished by adding two independent inhibitors, BADGP and OSMI1, in the last hour of PMA/ionomycin

restimulation. This treatment revealed that OGT inhibition significantly decreased the number of IFNγ-producing cells and reduced the overall ability of IFNγ production (Fig 5D–F). Thus, similar to inhibition of Th1 cell differentiation with 2-DG, inhibition of protein O-glycosylation reduces both, Th1 lineage commitment and effector function. In line, STAT1 phosphorylation on Tyr701 and Ser727 were significantly decreased after inhibition of OGT in the IFNγ+ population (Fig 5G and H), in line with reduction of O-glycosylation (Fig 5I). However, there were no observable changes in total STAT1 presence between the experimental groups (Fig 5J). These data emphasize that O-glycosylation mediated by

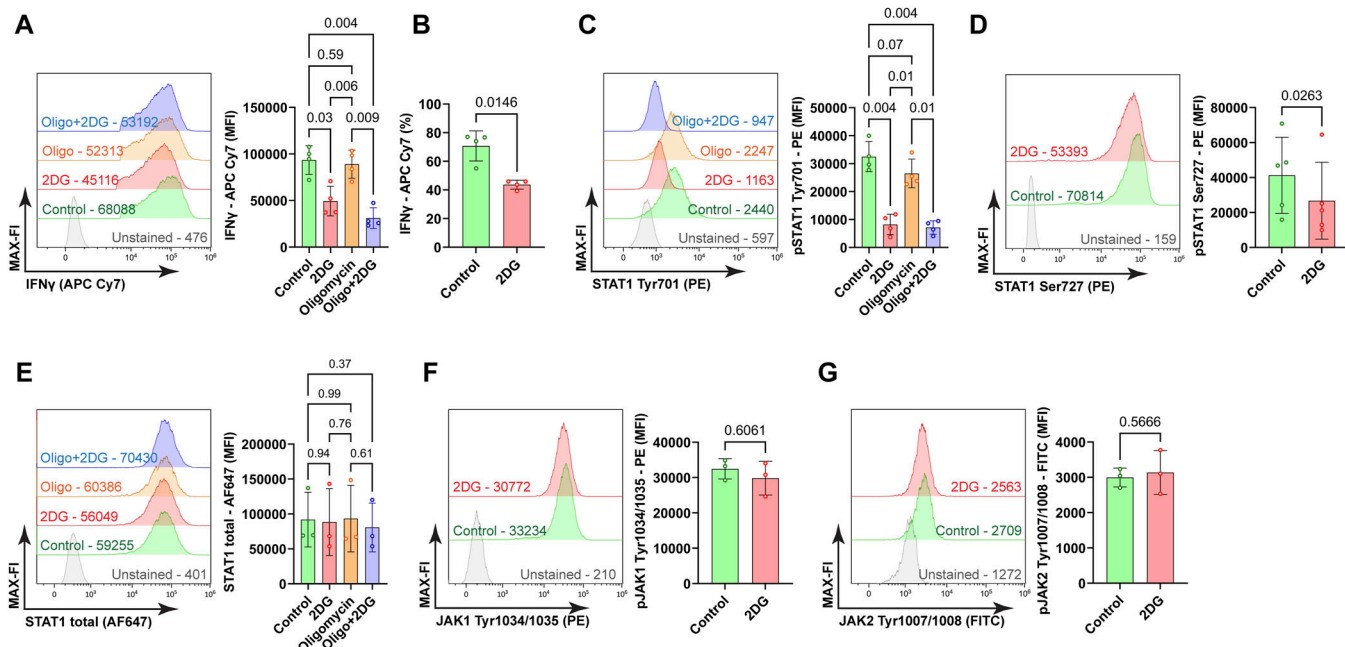

**Figure 3. Th1 identity and effector capacity depend on glycolysis.**
**(A)** Th1 cells were stimulated with PMA and ionomycin, treated for 30 min with 2-DG and oligomycin either separately or in combination and were subjected to intracellular staining for flow cytometry analysis. Representative histogram shows MFI (representative value indicated) of anti-IFNγ conjugated with APC Cy7. One-way ANOVA with repetitive measures, n = 4, mean ± SD. **(B)** Frequency of IFNγ+ cells, a Wilcoxon matched-pairs test, n = 4, mean ± SD. **(C)** Representative histogram showing the MFI of anti-pSTAT1 Tyr701 conjugated with PE gated on IFNγ+ cells. One-way ANOVA with repetitive measures, n = 4, mean ± SD. **(D)** Representative histogram depicting MFI of anti-pSTAT1 Ser727 conjugated with PE gated on IFNγ+ cells, two-sided paired t test, n = 5, mean ± SD. **(E)** Representative histogram showing MFI of anti-STAT1 antibody conjugated with Alexa Fluor 647 (AF647) gated on IFNγ+ cells, one-way ANOVA with repetitive measures, n = 3, mean ± SD. **(F)** Representative histogram, MFI of anti-JAK1 Tyr1034/1035 conjugated with PE gated on IFNγ+ cells, Wilcoxon matched-pairs test, n = 3, mean ± SD. **(G)** Representative histogram, MFI, of anti-JAK2 Tyr1007/1008 conjugated with FITC gated on IFNγ+ cells, Wilcoxon matched-pairs test, n = 3, mean ± SD.

OGT maintains Th1 cell identity and effector function by acting on STAT1 activity, indirectly or directly.

### Glycolysis- and O-glycosylation transferase mediate STAT1 O-glycosylation

The decrease of STAT1 phosphorylation in 2-DG-treated Th1 cells and in Th1 cells with OGT inhibition suggested an association between OGT activity and STAT1. To test whether O-glycosylation occurs on STAT1 itself in Th1 cells, we immunoprecipitated (IP) STAT1 from Act.T and Th1 cells, respectively, and analyzed STAT1 O-glycosylation by Western blot. There was more O-glycosylated STAT1 in Th1 cells when compared with Act.T cells (Fig 6A and B). Next, we tested whether inhibition of glycolysis and OGT impairs STAT1 O-glycosylation (Fig 6C and D). In fact, both inhibition of glycolysis with 2-DG, and OGT inhibition with BADGP and OSMI1 diminished STAT1 O-glycosylation. Together, these data support the hypothesis that OGT-mediated O-glycosylation of STAT1 itself drives Th1 commitment and function.

### STAT1 glycosylation is critical for lineage commitment

The previous experiment suggested that O-glycosylation of STAT1 itself instructs Th1 lineage commitment as well as Th1 effector function. To test for an intrinsic STAT1 effect that mediates glycolysis- and OGT-

driven Th1 effector identity and capacity, we analyzed the primary structure of STAT1. There are three predicted glycosylation sites, namely Thr489, Ser499, and Thr510 (https://glygen.org/protein/P42224#glycosylation). Whereas Thr489 O-glycosylation was identified once in Jurkat cells (Xu et al, 2022), Ser499 and Thr510 O-glycosylation of STAT1 were identified in eight different human cell lines and primary cell types (Hahne et al, 2013; Jitschin et al, 2019; Liu et al, 2020; Ramirez et al, 2021; Vang et al, 2024; Xie et al, 2021), most importantly also in activated human T cells (Lund et al, 2016; Woo et al, 2018). To validate the idea that intrinsic STAT1 glycosylation preserves Th1 lineage commitment and effector function, we replaced the two STAT1 glycosylation sites for which the best evidence existed, Ser499 and Thr510, with Alanine via CRISPR/Cas9 mediated integration of a homology directed repair (HDR) template. A WT (unedited) STAT1 sequence was used as a control and a linked knock-in of GFP was used to identify the edited cells (Fig 7A; Table S1). T cells were activated in two ways: first, similar to all the previous experiments, IL-12 was added directly after seeding to drive immediate Th1 cell differentiation. In this early differentiation setting, flow cytometry demonstrated reduced IFNγ abundance in STAT1 mutated (STAT1^mut; GFP+) versus unedited (GFP−) cells (Fig 7B; raw data in Table S2) as well as fewer Th1 cells (Fig 7C; raw data in Table S2). Whereas the reduction of IFNγ abundance was mild in most cases, it was observed in 9 out of 10 analyses. The edited GFP+ STAT1^wt cells did not show decreased IFNγ abundance (Fig 7D; raw data in Table S2)

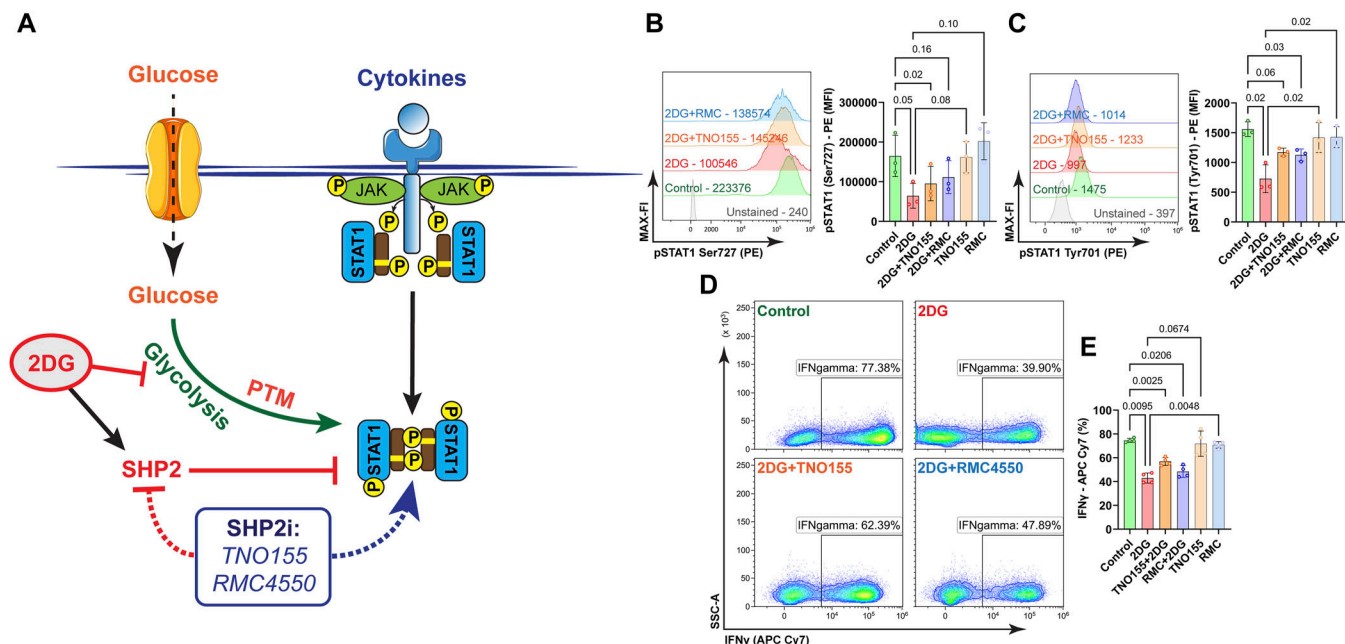

**Figure 4. Glycolysis supports STAT1 phosphorylation and IFNγ abundance independent of SHP2.**
**(A)** Schematic depicting the experimental setup and read-out. SHP2 inhibitor (SHP2i) posttranslational modification (PTM). PMA and ionomycin-stimulated Th1 cells were further treated with 2-DG, TNO155, and RMC4550 in combination and intracellularly stained for flow cytometry analysis. **(B)** Representative histogram showing representative median FI (MFI) of anti-pSTAT1 Ser727 conjugated with PE gated on IFNγ⁺ cells and analysis of pSTAT1-PE MFI, one-way ANOVA with repetitive measures, n = 3, mean ± SD. **(C, D)** Representative histogram with MFI of anti-pSTAT1 Tyr701 conjugated with PE from IFNγ⁺ cells and analysis of pSTAT1-PE MFI, one-way ANOVA with repetitive measures, n = 3, mean ± SD, (D) frequencies of IFNγ⁺, representative contour plots. **(E)** Frequencies of IFNγ⁺ cells, one-way ANOVA repetitive measures, n = 4, mean ± SD.

but also reduced Th1 cell differentiation (Fig 7E; raw data in Table S2), most likely due to transient *stat1* gene inactivity during the editing process. Therefore, these data suggest that Ser499 and Thr510 may not be involved in Th1 cell differentiation but do regulate Th1 effector capacity. To establish Ser499Ala and Thr510Ala mutations before the onset of Th1 cell differentiation, Th1 cell differentiation was induced only after CRISPR editing. Similarly, IFNγ expression was lowered in edited STAT1^mut cells during late induction of differentiation (four out of six experiments) whereas Th1 cell frequency was not affected (Fig S4). There was no effect on IFNγ production or Th1 cell frequency in edited STAT1^wt cells (Fig S4A–D), indicating a potentially specific effect of STAT1^mut on IFNγ abundance in this setup. In line with the effect of STAT1^mut on IFNγ generation, analysis of STAT1 in the IFNγ⁺ population showed a significant reduction of pSer727 in all the cells with point mutations (GFP⁺) (Figs 7F and S4E–H), irrespective of the differentiation protocol, whereas no change was observed at pTyr701 (Fig 7G). Moreover, edited STAT1^wt cells showed no reduction of pSer727 or pTyr701 (Fig 7H and I). Together, these data indicate that O-glycosylation of STAT1 selectively stabilizes or fosters phosphorylation of STAT1 at Ser727, and, thereby, sustains Th1 cell effector function as defined by the intracellular amount of IFNγ.

## Discussion

Mechanisms that intercalate metabolism and the signaling pathways that drive IFNγ production in primary human Th1 cells are scarcely reported. This study reveals that glycolytic flux and OGT activity both identify and determine Th1 lineage commitment as well as their effector function. Downstream of glycolytic flux, OGT promotes STAT1 phosphorylation at Tyr710 and Ser727 as well as its O-glycosylation. In accordance with reported O-glycosylation of STAT1 at Ser499 and Thr510 in primary activated human T cells (Lund et al, 2016; Woo et al, 2018), we find that these two residues selectively fine-tune STAT1 controlled IFNγ abundance and Ser727 phosphorylation. Thus, STAT1 O-glycosylation, at least at Ser499 and Thr510, is not involved in initial IL-12 driven Th1 cell polarization. We therefore devise a model in which glycolysis- and OGT-activity branch upstream of STAT1 O-glycosylation. Together, these findings reveal glycolysis's dual role in providing energy and substrates for posttranslational modifications, ensuring Th1 lineage stability. Because even pharmacological short-term inhibition of glycolysis- and OGT-activity influenced IFNγ production and STAT1 phosphorylation, we propose that this mechanism equips Th1 cells to adapt dynamically to their environment; this could have important consequences for anti-tumor immunity. This notion is strongly supported by very recent data showing reduced IFNγ production in GLUT1-deficient human T cells or human T cells treated with a GLUT-1 inhibitor (Chen et al, 2023; Jong et al, 2025), yet we provide mechanistic insights downstream of glucose uptake.

In addition to the descriptive confirmation of the well-established STAT1-IFNγ pathway in primary human T cells, we reinforced the relevance of this pathway in primary human T cells by Crispr/Cas9-mediated destruction of the *stat1* gene. Compared with the mock control, frequencies of IFNγ producing cells and IFNγ abundance were reduced in the *stat1* edited cells.

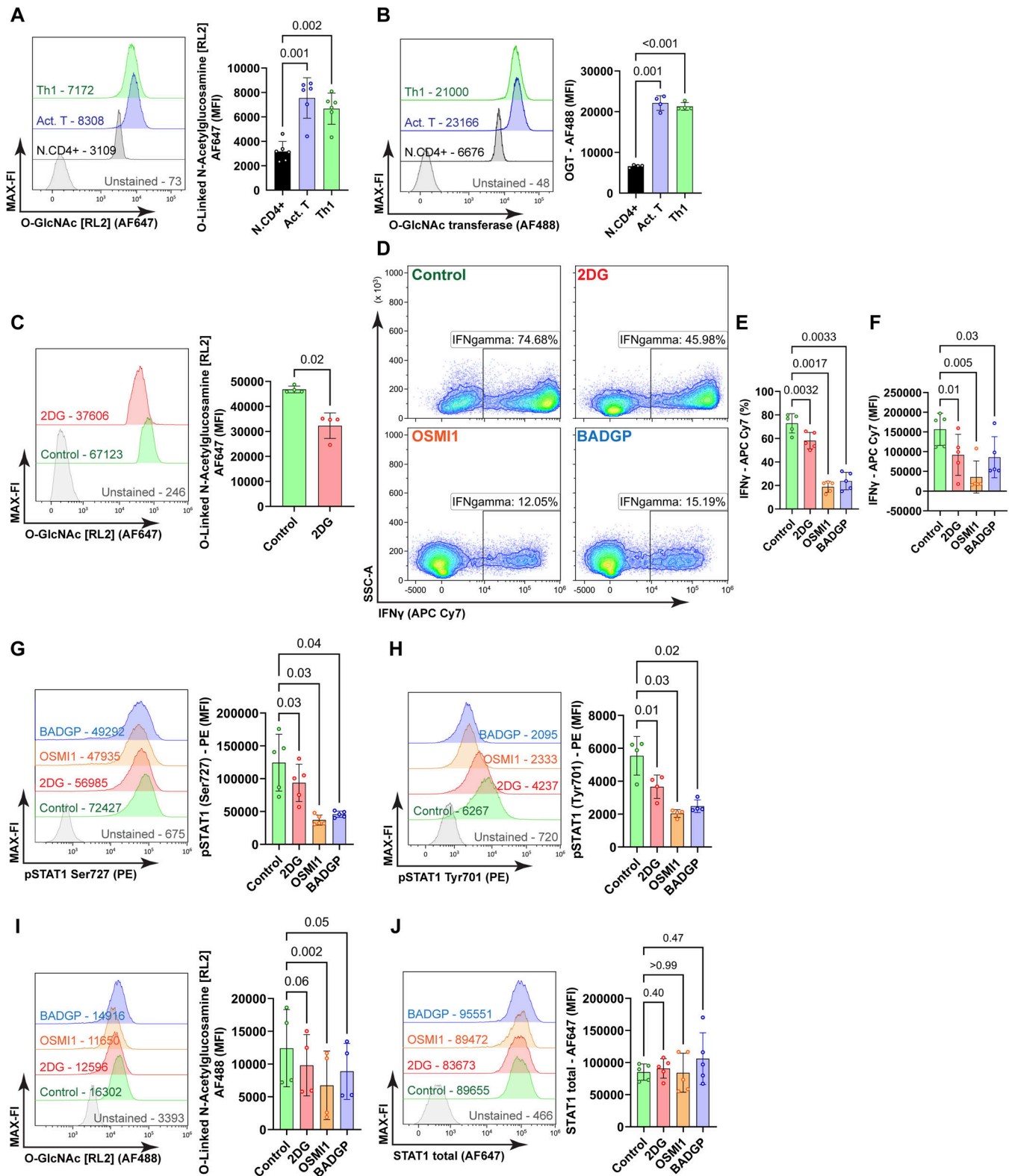

**Figure 5. O-glycosylation controls Th1 identity and effector capacity via STAT1 activity.**
**(A)** Total O-glycosylation, representative histograms with representative median fluorescence intensity (MFI) of anti-O-glycosylation antibody conjugated with Alexa Fluor 647 (AF647), one-way ANOVA with repetitive measures, n = 6, mean ± SD. **(B)** Representative histograms with MFI of the anti-O-glycosylation transferase antibody conjugated with Alexa Fluor 488 (AF488), one-way ANOVA with repetitive measures, n = 4, mean ± SD. **(C)** Representative histograms with MFI of the anti-O-glycosylation antibody conjugated with Alexa Fluor 488 (AF488) upon inhibition of glycolytic flux with 2DG. Wilcoxon matched-pairs test, n = 4, mean ± SD. **(D)** Frequencies of IFNγ[+] cells in control Th1 cells and 2-DG, OSMI1, and BADGP treated cells, representative contour plots. **(E)** Frequencies of IFNγ[+] cells, one-way ANOVA repetitive measures,

We provide several independent lines of evidence that dominant glycolytic metabolism distinguishes Th1 cells from Act.T cells, although Act.T cells are also glycolytic. Only a small difference was observed for the uptake of the glucose analog 6-NBDG. 6-NBDG exhibits a high binding affinity to GLUT1—~300-fold higher than glucose (Barros et al, 2009). Yet, there might be a receptor-independent uptake of 6-NBDG (Hamilton et al, 2021). In contrast, we have recently shown that 2-NBDG uptake does depend on the glucose transport GLUT1 (Bierling et al, 2024). The discrepancy might be explained by different cell types. We cannot exclude that T cells take up 6-NBDG in a receptor-independent manner but even if they did so, Th1 cells appear to use the glucose differently than Act.T cells as evidenced by increased acute NADH generation and lactate production. Functionally, inhibiting glycolysis for 30 min with 2-DG already reduced the frequency of IFNγ producing cells, IFNγ amount and STAT1 phosphorylation. To ensure a specific action of 2-DG on glycolysis, we corroborated these data with two additional inhibitors of glycolytic enzymes, IAA and OXA, acting on glyceraldehyde-3-phosphate-dehydrogenase and pyruvate kinase. Their effects mimicked 2-DG action but appeared in general to be smaller than that of 2-DG, although it is difficult to compare the efficacy of inhibitors acting on different enzymes. Nevertheless, it seems that the more downstream the inhibitors act in the glycolytic pathway, the less effective they are in suppressing IFNγ, STAT1 phosphorylation, and O-glycosylation. Downstream glycolysis inhibitors may be less effective in inhibiting glycolysis upstream of phosphofructokinase. Overall, these data are compatible with our interpretation that 2-DG likely exerts at least some of its inhibitory effect on Th1 cell differentiation by acting on the Hexosamine/OGT pathway. Notably, 2-DG did neither affect signaling upstream of STAT1, such as JAK1/2 activation, nor total STAT1 protein abundance. However, glycolysis inhibition by 2-DG did reduce STAT1 phosphorylation by acting on SHP2, which can dephosphorylate both Tyr710 and Ser727 (Wu et al, 2002; Chen et al, 2020). This notion is based on the fact that SHP2 inhibition with two different inhibitors partially restored STAT1 phosphorylation upon inhibition of glycolytic flux by 2-DG. These data strongly suggest that SHP2 activity is involved in regulating STAT1 activity via glycolysis, although we did not formally analyze SHP2 activity or exclude off-target effects of 2-DG on SHP2. These data therefore point to additional regulatory mechanisms of IFNγ production besides STAT1 tyrosine phosphorylation. In fact, glycolysis increased OGT expression and O-glycosylation in T.Act and Th1 cells and 2-DG reduced total O-glycosylation. These findings align with previous work showing increased OGT activity in activated primary human T cells (Lund et al, 2016).

OGT activity was required to maintain STAT1 O-glycosylation and phosphorylation as well as Th1 identity and IFNγ production. Thus,

OGT seems to be a major mediator of glycolytic pathways in T.Act and Th1 cells, and it has reportedly many substrates in primary T cells and Jurkat cells (Lund et al, 2016; Woo et al, 2018; Xu et al, 2022). Of note, protein O-glycosylation in contrast to protein N-glycosylation mainly targets cytoplasmic and nuclear proteins (Hart et al, 2007; Alfaro et al, 2012). O-glycosylation can strongly enhance the half-life of proteins in the nucleus, more so in the cytoplasm (Xu et al, 2022), and it can influence nuclear-cytoplasmic shuttling (Hart et al, 2007). For instance, O-glycosylation of nuclear factor κ B (NFκB) promotes its activation and nuclear translocation (Yang et al, 2008) whereas nuclear shuttling of nuclear factor of activated T cells (NFAT), another OGT substrate, is unaffected (Lund et al, 2016). On the other hand, lack of O-glycosylation reduces STAT5's tyrosine phosphorylation and transactivation potential (Freund et al, 2017). Consequently, O-glycosylation of transcription factors exerts protein-specific roles. This apparent specificity is in line with the very specific functions we deduce for the function of STAT1 O-glycosylation at Ser499 and Thr510, namely, maintenance of pSer727 and IFNγ amount. What could be further functional consequences of STAT1 O-glycosylation? Glycosylation can protect transcription factors from proteasomal degradation or dephosphorylation (Han & Kudlow, 1997; D. Liu et al, 2002). An increased half-life would be in agreement with findings for other proteins (Xu et al, 2022). Yet, our data in T cells do not indicate major effects of O-glycosylation on STAT1 protein half-life so far. It is possible that our experimental interventions were too short.

We extrapolate a proposed mechanism from CRISPR/Cas9 mediated gene editing. This was achieved by replacing Ser499 and Thr510 with alanine residues. Importantly, off-target effects of the gene editing process were controlled for by Ser499/Ser and Thr510/Thr replacements by the HDR template. Because O-glycosylation of STAT1 at Ser499 and Thr510 in primary human T cells is well established (Woo et al, 2018), we did not confirm the O-glycosylation of these particular residues. Still, we did corroborate that OGT targets STAT1 in Act.T cells, even more so in Th1 cells. It was surprising to see that Ser499 and Thr510 editing merely affect Ser727 phosphorylation and not Tyr510 phosphorylation. This specificity represents an important signaling loop for Th1 identity controlled by glycolytic flux. There may remain a putative role of Thr489 O-glycosylation. However, in stark contrast to Ser499 and Thr510, O-glycosylation of Thr489 was only found once at all, in Jurkat cells (Xu et al, 2022) and never in primary human T cells (Lund et al, 2016; Woo et al, 2018) or other cell types (https://glygen.org/protein/P42224#glycosylation).

The mechanism that we describe here may imprint inflammatory priming of Act.T cells by IL-12 and, consequently, present a potential therapeutic target that may help limit inflammation or boost anti-viral or anti-tumor effector function. This suggestion aligns with the inflammatory priming in MSC in which a similar mechanism was described, albeit, in MSC,

---

n = 5, mean ± SD. **(F)** MFI of IFNγ⁺, one-way ANOVA repetitive measures, n = 5, mean ± SD. **(G)** Representative histograms with MFI of the anti-pSTAT1 Ser727 antibody conjugated with PE, one-way ANOVA with repetitive measures, n = 5, mean ± SD. **(H)** Representative histograms with MFI of the anti-pSTAT1 Tyr701 antibody conjugated with PE, one-way ANOVA with repetitive measures, n = 4, mean ± SD. **(I)** Representative histograms with MFI of the anti-O-Glycosylation antibody conjugated with AF647, one-way ANOVA with repetitive measures, n = 4, mean ± SD. **(J)** A separate group of cells from the same setup was intracellularly stained with the anti-STAT1 antibody conjugated with Alexa Fluor 647 (AF647). Representative histograms with MFI, one-way ANOVA with repetitive measures, n = 5, mean ± SD.

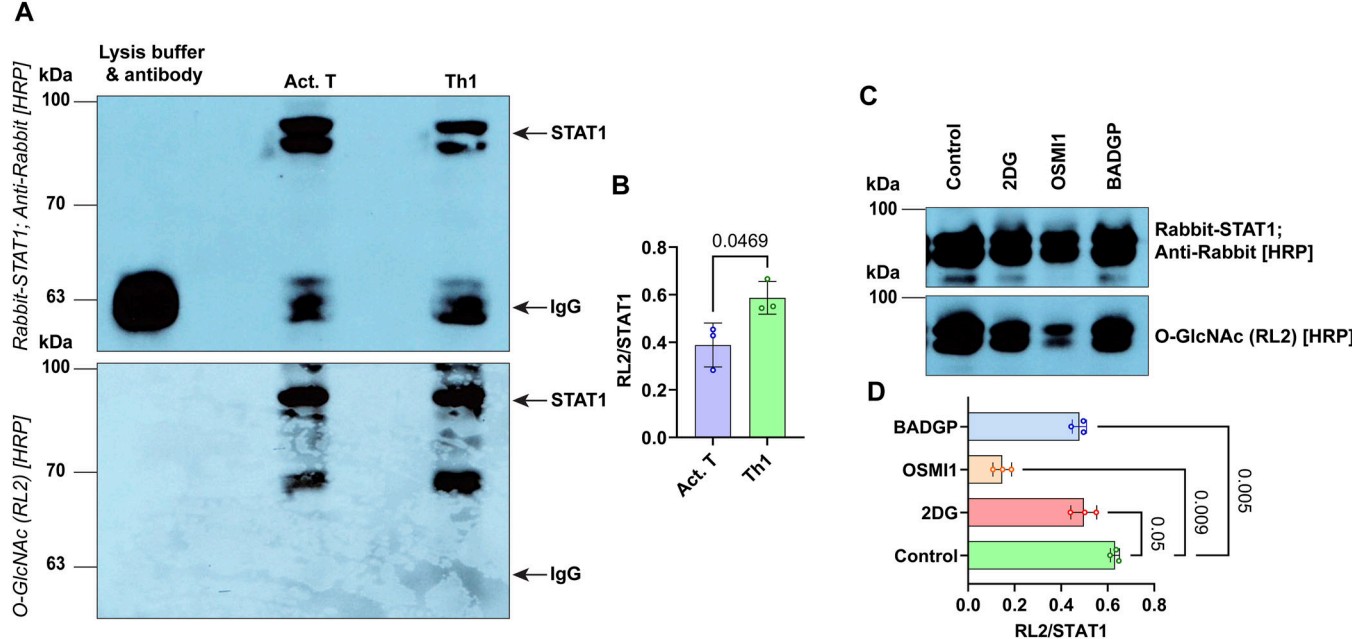

**Figure 6. Analysis of STAT1 O-glycosylation.**
**(A)** Cells were harvested after PMA/ionomycin stimulation, lysed, and STAT1 was immunoprecipitated (IP). Subsequently, samples were separated by 10% SDS–PAGE, followed by Western blot. The membrane was stained with the horseradish peroxidase (HRP) conjugated anti-O-glycosylation (RL2) antibody, washed and restained with a rabbit anti-STAT1 antibody, followed by anti-rabbit-HRP. The position of the STAT1 band and IgG are depicted. **(B)** quantification of the intensity of the RL2/STAT1 signal ratio after scanning, two-sided paired $t$ test, n = 3, mean ± SD. **(C)** Th1 cells were treated with 2-DG, OSMI1, and BADGP and were subjected to STAT1 IP, followed by Western blot and membrane staining with STAT1 and RL2 detection. **(D)** The intensity of the RL2/STAT1 signal ratio was calculated among the control (Th1), 2-DG, OSMI1, and BADGP treated groups, one-way ANOVA with repetitive measures, n = 3, mean ± SD. Molecular mass standards (kD) on the left.
Source data are available for this figure.

STAT1 glycosylation elicits an immunoregulatory phenotype (Jitschin et al, 2019). Because the mechanism of glycolysis- and OGT-driven STAT1 glycosylation appears to be shared in human MSC and Th1 cells, this signaling circuit might be a general and conserved pathway, yet with cell type specific outcomes. Future experiments should aim at identifying O-glycosylation dependent STAT1 target genes in different cell types. Our study underscores the previous finding that STAT1 O-glycosylation integrates nutrient sensing with immunoregulatory processes in MSCs, forming a mechanism for controlling inflammation (Jitschin et al, 2019). IL-12 is crucial for Th1 cell polarization and its abundance as well as polymorphisms of IL-12 and its receptor are linked to various inflammatory diseases (Tan et al, 2009). Along this line, overnutrition, such as hyperglycemia, can support inflammation as well. It is tempting to speculate that continuous availability of glucose creates a self-sustaining pro-inflammatory loop via enhanced glycolysis and O-glycosylation in Th1 cells. Of note, glucose and other nutrients also induce the mTOR pathway. Interestingly, there is crosstalk between mTOR and STAT signaling (Ramana et al, 2002; Kristof et al, 2003; Kroczynska et al, 2009; Jitschin et al, 2019). The specific mode of interaction between mTOR and STAT1 signaling pathways requires further investigation but STAT1 O-glycosylation might contribute to the interplay, albeit this is speculative at the moment. In conclusion, our observations demonstrate a mechanistic connection between glycolysis, STAT1 phosphorylation, and its O-glycosylation during human Th1 differentiation and effector capacity.

# Material and Methods

## Reagents, kits, and software

See Tables S3, S4, S5, S6, S7, and S8.

## Human PBMCs

PBMCs were freshly isolated via density gradient centrifugation from leukoreduction system chambers (LRSCs) of healthy human donors. The permission to use this LRSC from the Department of Transfusion Medicine and Haemostaseology at Universitätsklinikum Erlangen was given by the Ethics Committee of the Friedrich-Alexander-Universität Erlangen-Nürnberg (ethical approval no. 48_19 B and 21-400-Bp). The blood was diluted 1:1 with PBS and added to a tube containing 15 ml of Pancoll human (density: 1.077 g/ml). After centrifugation, the PBMCs were collected from the interface of the Pancoll/erythrocyte mixture and serum. The PBMCs were cryopreserved in the vapor phase of liquid nitrogen. The samples were subsequently thawed and used for further experiments.

## Naïve CD4[+] T cell isolation

Naïve CD4[+]CD45RA[+] T cells were isolated from thawed PBMC with the Human Naïve CD4[+] T Cell Isolation Kit II (from Miltenyi Biotec, at.

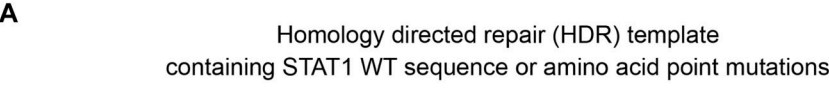

**Figure 7. Analysis of Th1 effector function and STAT1 Phosphorylation upon mutation of the STAT1 O-glycosylation sites.**
**(A)** Schematic of the Stat1 gene editing approach, LHA, left homology arm, RHA, right homology arm, representative histogram of GFP⁺ or GFP⁻ electroporated cells.
**(B)** Representative contour plots of IFNγ producing cells show the median fluorescence intensity (MFI; representative value indicated) of the anti-IFNγ antibody conjugated with APC Cy7, and the population was gated from GFP⁺ or GFP⁻ population from STAT1mut cells (early differentiation). The statistical analysis was performed with a Wilcoxon matched-pairs test, $P = 0.0250$, n = 10. **(C)** The statistical analysis of IFNγ producing cells was performed with a two-sided paired $t$ test, $P = 0.0097$, n = 10. **(D)** MFI of GFP⁺IFNγ⁺ and GFP⁻IFNγ⁺ cells from the STAT1wt early differentiation group were statistically analyzed with a Wilcoxon matched-pairs test, n = 09. **(E)** Representative contour plots of IFNγ producing cells gated from GFP⁺ or GFP⁻ population from STAT1wt cells (early differentiation). The statistical analysis was performed with a two-sided paired $t$ test, $P = 0.0042$, n = 9. **(F)** Representative histogram shows the MFI of the anti-STAT1 Ser727 antibody conjugated with PE gated from GFP⁺IFNγ⁺ or GFP⁻IFNγ⁺ population from STAT1mut cells (early differentiation). The statistical analysis was performed with a Wilcoxon matched-pairs test, $P = 0.0187$, n = 5. **(G)** MFI of STAT1 Tyr701 (PE) of GFP⁺IFNγ⁺ and GFP⁻IFNγ⁺ cells from the STAT1mut early differentiation group were statistically analyzed with a two-sided paired $t$ test, $P = 0.4710$, n = 5. **(H)** MFI of STAT1 Ser727 (PE) of GFP⁺IFNγ⁺ and GFP⁻IFNγ⁺ cells from the STAT1wt early differentiation group were statistically analyzed with a two-sided paired $t$ test, $P = 0.0625$, n = 5. **(I)** MFI of STAT1 Tyr701 (PE) of GFP⁺IFNγ⁺ and GFP⁻IFNγ⁺ cells from the STAT1wt early differentiation group were statistically analyzed with a two-sided paired $t$ test, $P = 0.5920$, n = 5.

No.: 130-094-131) via magnetic cell sorting (MACS). In brief, PBMC were washed with the MACS buffer (PBS, pH 7.2, 0.5% human serum albumin [HAS], and 2 mM ethylenediaminetetraacetic acid or EDTA).10 μl of MACS Naive CD4⁺ T Cell Biotin-Antibody Cocktail II, human, were added per 10⁷ cells and incubated for 5 min at 4°C. Subsequently, 20 μl of Naive CD4⁺ T Cell MicroBead Cocktail II were added per 10⁷ total cells, followed by the addition of some MACS buffer. After a 10-min incubation period, the cells were passed through a MACS LS column (Cat. No.: 130-042-401; Miltenyi Biotec) and placed into a MACS magnet system (Cat. No.: 130-091-051; QuadroMACS Separator; Miltenyi Biotec). Both the column and the magnet system were precooled to 4°C. The isolated untouched

cells were collected into a Falcon tube for subsequent culture. A small sample of cells was analyzed to determine the purity of the CD4+CD45RA+ population via flow cytometry. The accepted purity for the Th1 culture system was between 96% and 98%.

### Cell culture

The culture was set up in a 96-well plate coated with Ultra-LEAF anti-CD3 (3 $\mu$g/ml) (Cat. No.: 300438; BioLegend) and anti-CD28 (5 $\mu$g/ml) (Cat. No.: 302934; BioLegend) antibodies in 50 $\mu$l PBS per well. After a 2-h incubation period at 37°C, the plates were aspirated, and cells were seeded at a density of 50,000 cells/well in a volume of 200 $\mu$l. Standard R10 medium was prepared and used for the culture by combining 500 ml of RPMI1640 (Cat. No.: 11875093; Gibco, Thermo Fisher Scientific) with 10% heat-inactivated FCS, 2 mM L-glutamine, 50 U/ml penicillin G, 50 $\mu$g/ml streptomycin, and 50 $\mu$M $\beta$-mercaptoethanol. Activated and Th1 differentiation cultures were supplemented with 8 ng/ml IL-2 (Cat. No.:130-097-743; Miltenyi Biotec) and 10 $\mu$g/ml anti-IL-4 (Cat. No.:500802; BioLegend). Furthermore, the Th1 cultures were supplemented with 8 ng/ml IL-12, to promote differentiation. After a 48-h incubation period at 37°C with 5% $CO_2$, the Th1 culture was transferred to a fresh plate. After an additional 72-h incubation period, the cells were restimulated with 10 ng/ml PMA (Cat. No.:ab120297; Abcam) and 500 ng/ml ionomycin (Cat. No.: I3909; Sigma-Aldrich) with 1 $\mu$l/ml GolgiPlug (Cat. No.: 555029; BD Biosciences) for 4 h before flow cytometry analysis. In some experiments, cells were treated with BADGP (2.5 mM for 1 hr) and OSMI1 (40 $\mu$M for 1 h) (Jitschin et al, 2019; Y. Liu et al, 2017), iodoacetate (IAA, 100 mM for 30 min), and oxalate (OXA, 1 mM for 30 min) (Ho et al, 2015). Unless otherwise indicated, this culture system was used for all experiments.

### Flow cytometry

Approximately 2 × 10[5] cells were collected and rinsed with cold PBS. Subsequently, the cells were stained with Zombie Aqua (1:1,000; Fixable Viability Kit; Cat. No.: 423101; BioLegend) for 15 min at RT in the dark. Then, the cells were washed twice with PBS and 400 $\mu$l of CytoFix buffer (preheated at 37°C; Cat. No.: 554655; BD Cytofix Fixation Buffer; BD) was added to each sample, followed by a 10-min incubation at 37°C. After a wash with cold PBS, the cells were permeabilized using PhosFlow buffer (precooled at −20°C; Cat. No.: 558050; BD Phosflow Perm Buffer III; BD) and incubated at 4°C for 30 min. After that the cells were washed twice with FACS buffer (PBS + 0.25% BSA) and incubated for 10 min with BD Fc Block (BD, Cat. No.: 564219, dilution: 1:100). The cells were then stained with the corresponding antibodies (detailed information is in the supplemental section): Panel 1, analysis of cell purity after MACS isolation (APC anti-human CD4, 1:100, FITC anti-human CD45RA, 1:100); panel 2, analysis of Th1 differentiation (FITC anti-human IL-17A, 1:20, FITC anti-human IL-17A, 1:20, PE mouse Anti-Stat1 [pTyr701] [1:5] or PE anti-STAT1 phospho Ser727 [1:20], PerCP/Cyanine5.5 anti-T-bet [1:50], Alexa Fluor 647 mouse anti-total Stat1 [N-Terminus], 1:5, APC/Cyanine7 anti-human IFN-γ, 1:25, Brilliant Violet 421 anti-human IL-4, 1:20, Zombie Aqua, 1:1,000); panel 3, analysis of

STAT1 upstream signaling (FITC phospho-Jak2 [Tyr1007, Tyr1008] recombinant rabbit monoclonal, 1:20, PE phospho-Jak1 [Tyr1022, Tyr1023] recombinant rabbit monoclonal, 1:20, PerCP/Cyanine5.5 anti-T-bet, 1:5, APC/Cyanine7 anti-human IFN-γ, 1:25, Zombie Aqua, 1:1,000); panel 4, analysis of O-Glycosylation (Alexa Fluor 488 anti-O-glycosylation monoclonal [RL2] [1:20] or Alexa Fluor 647 anti-O-glycosylation [RL2] [1:100], FITC anti-OGT1 polyclonal, 1:100, PE mouse anti-Stat1 [pTyr701] [1:5] or PE anti-STAT1 phospho Ser727 [1:20], PerCP/Cyanine5.5 anti-T-bet, 1:50, APC/Cyanine7 anti-human IFN-γ, 1:25, Zombie Aqua, 1:1,000). After a 30-min incubation period with the antibodies, the cells were washed twice. The flow cytometry analysis was conducted using a Beckman Coulter Gallios Flow Cytometry system, and the resulting data were analyzed using the Beckman Coulter Kaluza Analysis Software.

### SCENITH: single cell ENergetIc metabolism by profiling translation inhibition

The SCENITH experiment was conducted as previously described (Argüello et al, 2020), with minor modifications. In summary, ~1 × 10[6] naïve CD4+, activated T, and Th1 cells were restimulated with phorbol 12-myristate 13-acetate (PMA) and ionomycin for 4 h. During the final hour of the experiment, cells were treated with 2-deoxy-D-glucose (100 mM) and oligomycin (1 $\mu$M), either separately or in combination, for ~45 min. In the final 15 min of the experiment, puromycin was added at a concentration of 10 $\mu$g/ml. After the conclusion of the treatment period, the cells were washed with cold PBS and stained using an Fc receptor blockade and Zombie Aqua as the death cell marker. After another wash, the cells were fixed, permeabilized, and subjected to intracellular staining as previously described, incorporating an anti-puromycin monoclonal antibody conjugated with Alexa Fluor 488 (Cat. No.: 381506; 1:200; BioLegend).

### Glucose uptake assay

A total of ~10[6] cells were harvested and washed with FACS buffer. Subsequently, the cell pellets were reconstituted in a standard R10 medium prepared with glucose-free RPMI1640, and 300 $\mu$M of 6-NBDG was introduced to the samples. After a 30-min incubation period at 37°C, the samples were washed again with FACS buffer in preparation for flow cytometry analysis.

### Extracellular flux analysis

Extracellular flux analysis was performed using a Seahorse XFe96 device following the previously published protocol (Abir et al, 2024) and the Seahorse protocol provided by Agilent Inc. In summary, After PMA/ionomycin restimulation, cells were collected and washed with their corresponding Glyco-stress test (GST), Mito-stress test (MST), and the ATP analysis medium. About 1.8 × 10[5] cells were seeded to the Seahorse 96-well cell culture plate coated with Poly-D-Lysine and after 45 min of incubation in a $CO_2$-free incubator, they were subjected to the experiment. The data were analyzed with the Agilent Wave software.

## Glucose and lactate measurement

The supernatant from cultures of naïve CD4[+] cells, activated T cells, and Th1 cells were collected on the last day. The glucose and lactate levels in the medium were determined using the Super GL compact instrument (Hitado GmBH), as described previously (Urbanczyk et al, 2022). Before the introduction of the samples into the measuring Eppendorf, the instrument was calibrated with Super GL calibration fluid. With the help of a capillary, the test sample was introduced to the Glucocapil reaction vessels, which were then introduced to the device. Subsequently, the device automatically calculated and displayed the results.

## Immunoprecipitation (IP) and Western blot

Approximately $10^7$ viable cells were lysed in NP40 cell lysis buffer (1% Nonidet P-40, 150 mM NaCl, 5 mM EDTA, 50 mM Tris/HCl, pH 7.4) containing 1 mM phenyl-methyl-sulfonyl-fluoride (PMSF) and 100 mM NaVO$_3$. The cells were lysed on ice for 10 min, followed by a 10-min centrifugation at 10,000$g$ and 4°C. Supernatants were collected and incubated with 5 $\mu$l of mouse $\alpha$STAT1 and 50 $\mu$l of Protein G Sepharose for 2 h at 4°C on a mixing rotor. The control sample was prepared by incubating 5 $\mu$l of the antibody with 50 $\mu$l Protein G Sepharose, in buffer only. Beads were washed twice with NP40 buffer and boiled at 65°C for 5 min in 1 x SDS-sample loading buffer (2% SDS, 62.5 mM Tris/HCl, pH 6.8, 10% glycerol, 100 mM dithio-threitol). Samples were subjected to 10% SDS–PAGE and Western blotting to nitrocellulose for 45 min at 400 mA and 25 V. The membrane was stained with Ponceau S for 2 min and rinsed with Tris-buffered saline (50 mM Tris/HCl, pH 7.4, 150 mM NaCl) including 0.1% Tween-20 (TBST) for 15 min. Then the membrane was blocked with 5% skimmed milk in TBST for an hour. The membrane was washed again for 30 min and stained with primary and secondary antibodies for an hour, followed by 4 × 5 min washes with TBST. The membrane was developed with an in house enhanced chemiluminescence (ECL) kit for 2 min and exposed to X-ray films. Developed X-ray films were scanned, converted to TIFF and analyzed with ImageJ.

## Homology directed repair (HDR) DNA template

The HDR templates were synthesized by Twist Bioscience, CA. The left homology arm (LHA; 386 bp) is followed by the WT sequence of STAT1 exon 18 or an edited sequence containing the amino acid point mutations S499A and T510A. The subsequent self-cleaving peptide P2A separates the *stat1* sequence from an EGFP reporter. After the stop codon (TAA), the 376 bp right homology arm (RHA) concludes the HDR template (Fig 7). The sequences of these segments are depicted in Table S1. The DNA construct was delivered as a sequence-verified plasmid. The lyophilized plasmid was reconstituted with sterile water to 100 ng/$\mu$l and amplified by PCR to generate a linearized double-stranded HDR template. Each 50 $\mu$l PCR reaction contained 5 x Q5 Reaction Buffer, 0.5 $\mu$M STAT1 HDR genomic forward primer targeting LHA (5'-AGAGGTGAAACAGGAAGCGAG-3'), 0.5 $\mu$M STAT1 HDR genomic reverse primer targeting RHA (5'-

CTTTCCCTTGGGAATTCATCTCAG-3'), 0.2 mM dNTPs, 0.5 $\mu$l Q5 DNA polymerase, and 200 ng reconstituted DNA in PCR grade water. The PCR was run with the following cycling conditions: Initial denaturation at 98°C for 5 min, 30 cycles of 98°C for 10 sec, 60°C for 30 sec, and 72°C for 3 sec, final elongation at 72°C for 2 min, and hold at 4°C. Successful amplification was confirmed with a 1% agarose gel, and the amplified HDR template was purified with a MinElute PCR Purification Kit (28004; QIAGEN) according to the manufacturer's instructions.

## Ribonucleoprotein (RNP) production

In brief, 40 $\mu$M gRNAs were produced by mixing equimolar amounts of trans-activating crRNA (tracrRNA) (1072534; Integrated DNA Technologies) with STAT1 crRNA (5'-GACTCCACCATGTGCACGAT-3'; Integrated DNA Technologies), incubated at 95°C for 5 min with subsequent cool down to RT. Afterward, 50 $\mu$g/sample poly-L-glutamic acid (PGA, P4761; Sigma-Aldrich) (1,2) and 20 $\mu$M electroporation enhancer (10007805; Integrated DNA Technologies) were added to the gRNA. RNP production was concluded by adding an equal volume of Cas9 Nuclease V3 (1081059, diluted to 6 $\mu$M; Integrated DNA Technologies) to the gRNA (40 $\mu$M), respectively. RNPs were incubated for 15 min at RT and subsequently stored on ice for processing on the same day (Nguyen et al, 2020; Kath et al, 2022).

## Electroporation and cultivation of edited cells

Naïve CD4[+]CD45RA[+] cells were cultured in differentiation medium (complete RPMI1640, IL-2, anti-IL-4 antibody and IL-12) for early differentiation or in activation medium (complete RPMI1640, IL-2, anti-IL-4 antibody) for late differentiation for 24 h and subjected to electroporation. Before electroporation, the DNA-sensing inhibitor RU.521 (inh-ru521; InvivoGen) was added to the cells at a final concentration of 4.82 nM for 6 h. Afterward, activation was stopped by transferring cells to a new plate in fresh complete RPMI medium. For electroporation (Shy et al, 2023), $1 \times 10^6$ activated cells per electroporation sample were resuspended in 20 $\mu$l P3 electroporation buffer (V4SP-3960; Lonza) and then mixed with DNA/RNP mix (0.5 $\mu$g HDR template, 3.5 $\mu$l STAT1 RNPs). After transfer into the 16-well Nucleocuvette Strip (V4SP- 3960; Lonza), cells were electroporated (pulse sequence EH100) in the Lonza 4D-Nucleofector. After electroporation, cells were seeded in 900 $\mu$l of antibiotic-free RPMI medium supplemented with 180 U/ml IL-2. After 15 min, 20 $\mu$l of a mixture containing 0.5 $\mu$M HDAC class I/II Inhibitor Trichostatin A (M1753; AbMole) and 10 $\mu$M DNA-dependent protein kinase (DNA-PK) inhibitor M3814 (CT- M3814; chemietek) were added to each sample. Cells were incubated for 12–18 h in 24-well plates. Aferward, cells were incubated in complete RPMI1640, IL-2, anti-IL-4 antibody and IL-12. GFP[+] cells were identified as successfully edited cells. Cells edited with Ser499 and Thr510 mutated constructs are marked as STAT1[mut], and WT constructs are marked as STAT1[wt]. Experiments were performed with three healthy donor samples with 3–10 technical replicates. Flow cytometry analysis was performed to compare GFP[+]IFNγ[+] cells with GFP[−]IFNγ[+].

## Statistical analysis

GraphPad Prism 9 was used for the statistical analysis (GraphPad Software). Data were analyzed by the Wilcoxon matched-pairs *t* test or the two-sided paired *t* test based on the data distribution. One-way or two-way ANOVA was used for comparisons involving more than two groups or factors under the assumption of approximately normal distribution and equal variances, with Tukey's or Dunnett's multiple comparisons tests to determine specific group differences with type I error control.

# Data Availability

Original data are available upon reasonable request. Please contact dirk.mielenz@fau.de.

# Supplementary Information

# Acknowledgements

We acknowledge all voluntary healthy donors. We thank Profs. Hyun-Dong Chan and Christina Zielinski for helpful discussions. The study was funded by the Deutsche Forschungsgemeinschaft (DFG) through the Research Training Grant (RTG) 2599, P02 (to D Mielenz) and P08 (to D Mougiakakos), FOR2886 "PANDORA," P03, to D Mielenz and G Krönke, a Start-up grant through the RTG2599 (to AH Abir), FOR5560, P08, to D Mielenz. K Schober is mainly supported by the German Federal Ministry of Education and Research (BMBF, project 01KI2013) and J Benz by the Interdisciplinary Center for Clinical Research (IZKF) at the University Hospital of the University of Erlangen-Nürnberg (project A98).

## Author Contributions

AH Abir: conceptualization, data curation, software, formal analysis, validation, investigation, visualization, methodology, project administration, and writing—original draft, review, and editing.
J Benz: conceptualization, data curation, formal analysis, methodology, and writing—original draft.
B Frey: resources, data curation, formal analysis, methodology, and writing—original draft and project administration.
H Bruns: resources, formal analysis, methodology, and project administration.
G Krönke: resources, supervision, funding acquisition, and writing—review and editing.
US Gaipl: resources, data curation, formal analysis, methodology, and project administration.
K Schober: conceptualization, resources, supervision, funding acquisition, methodology, project administration, and writing—original draft, review, and editing.
D Mougiakakos: conceptualization, supervision, funding acquisition, and writing—review and editing.
D Mielenz: conceptualization, resources, formal analysis, supervision, funding acquisition, validation, investigation, visualization, methodology, project administration, and writing—original draft, review, and editing.

## Conflict of Interest Statement

The authors declare that they have no conflict of interest.

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
