## [Reviewer comments · Life Science Alliance]

Glycolytic flux sustains human Th1 identity and effector function via STAT1 glycosylation

Ariful Haque Abir, Julia Benz, Benjamin Frey, Heiko Bruns, Gerhard Krönke, Udo Gaipl, Kilian Schober, Dimitrios Mougiakakos, and Dirk Mielenz

DOI: <https://doi.org/10.26508/lsa.202503315>

Corresponding author(s): Dirk Mielenz, Friedrich-Alexander-Universität Erlangen-Nürnberg

Review Timeline:

Submission Date:	2025-03-19
Editorial Decision:	2025-05-15
Revision Received:	2025-08-11
Editorial Decision:	2025-10-13
Revision Received:	2025-10-20
Accepted:	2025-10-21

Scientific Editor: Sarita Hebbar

Transaction Report:

May 14, 2025

Re: Life Science Alliance manuscript #LSA-2025-03315-T

Prof. Dirk Mielenz
Friedrich-Alexander-University of Erlangen-Nuremberg
Division of Molecular Immunology
Glueckstr. 6
Erlangen, Bavaria 91054
Germany

Dear Dr. Mielenz,

Thank you for submitting your manuscript entitled "Glycolytic flux sustains human Th1 identity and effector function via STAT1 glycosylation" to Life Science Alliance. The manuscript was assessed by three expert reviewers, whose comments are appended to this letter.

The three reviewers noted that this work is of potential value to the field. That said the reviewers raised significant concerns that must be addressed before publication at LSA. Addressing these concerns might require new experiments and/or re-analyses of existing data, and we recommend that the following must be included:

1. STAT1 Phosphorylation upon Th1 differentiation:

Provide data for total STAT1 levels (Reviewer 2, point 1)

2. Metabolic state of Th1 cells:

Provide the data for ATP calculation and OCR curves (Reviewer 2, point 3 and Reviewer 3, point 2) .

3. Use of 2DG as a glycolytic inhibitor:

Re-analyses and/or data description for the role of glycolysis and glucose uptake (Reviewer 2, points 4- 7, Reviewer 3, point 2). You may choose to not remove the experiments with fluorescent glucose analogue 6-NBDG, in which case please discuss the limitations as pointed out by Reviewer 2.

Experiment using one of the additional glycolysis inhibitors (Reviewer 3, point 4).

4. SHP2 inhibition:

Experiments show the effects of the SHP2 inhibitors in isolation and compare it to 2DG treatment alone (Reviewer 1, paragraph 4 and Reviewer 2, point 8).

In line with the overall evaluation of the reviewers, we invite you to submit a revised manuscript addressing their comments. When submitting the revision, please include a letter addressing the reviewers' comments point by point. While a rebuttal must respond to all points in some form, additional data to resolve these points (other than ones indicated above) is not required. The typical timeframe for revisions is three months. Please note that papers are generally considered through only one revision cycle, so strong support from the referees on the revised version is needed for acceptance.

Thank you for this interesting contribution to Life Science Alliance. We are looking forward to receiving your revised manuscript.

Sincerely,

Sarita Hebbar, PhD
Scientific Editor
Life Science Alliance
<http://www.lsjournal.org>

B. MANUSCRIPT ORGANIZATION AND FORMATTING:

Reviewer #1 (Comments to the Authors (Required)):

In the submitted manuscript, Ariful Haque Abir et al. investigated the effect of glycolysis in regulation of human Th1 cell identity and effector function. Their data showed that inhibition of glycolysis reduced STAT1 phosphorylation and IFN γ production, which is not completely dependent on JAK1/2 and SHP2 activity. Instead, they demonstrated that regulation of human Th1 differentiation by glycolysis is via STAT1 O-Glycosylation. As such, the study improves our understanding of mechanisms which drive human Th1 cell differentiation and IFN γ production. However, the following concerns need to be addressed before the study is published.

Like the authors correctly indicated in "Introduction", page 4 line 111, "STAT1 is essential for Th1 cell differentiation and the production of IFN γ ". This has been well demonstrated both in mice and human T cells. Therefore, Figure 1 does not provide novel findings or novel information. As the authors described (page 6 line 160) "These data validate efficient human Th1 cell differentiation in our cultured cells", this figure should be presented as supplementary figure. The texts should be simplified accordingly.

The authors concluded that "2-DG-induced STAT1 dephosphorylation partially depends on SHP2" (page 12 line 320), 2-DG may intrinsically activate SHP2. They showed that "SHP2 inhibition elevated the phosphorylation" (page 13 line 329). Since the inhibitor may have other effects, they should have shown by western blot the SHP2 activation, e.g. by detecting the pSHP2.

In Figure 4b, c, d, statistical analysis of the two SHP2 inhibition versus 2-DG should be included.

As one of the most important conclusions, the authors claimed that OGT inhibition significantly decreased IFN producing cells in Figure 5d-f). What concentrations of the two inhibitors BADGP and OSMI1 were used? The authors should present data showing the reduced IFN producing cells are not due to the increased cell death? Furthermore, the authors should provide data showing the dose dependent effect of these two inhibitors.

In Figure 7, how were the p values calculated? The information on statistical methods used in each of the comparisons is missing. Please add it. Please provide the raw data of each IFN MFI and percentage in Fig7b---i on a supplementary table.

Reviewer #2 (Comments to the Authors (Required)):

In this manuscript Abir et al describe metabolic features of human CD4+ T cell populations, highlighting that in vitro generated Th1 cells demonstrate elevated glycolytic capacity, which appears to functionally underpin their capacity to express the hallmark cytokine interferon-gamma (IFN-gamma). Mechanistically, this relates to increased O-glycosylation of the signalling molecule STAT1 since pharmacological O-glycosyltransferase inhibition or CRISPR/Cas9-mediated mutation of STAT1 glycosylation sites diminish STAT1 phosphorylation and IFN-gamma expression. This is an interesting study highlighting a novel mechanism whereby T cell metabolic activity impacts their differentiation and function. However, I have a number of concerns that should be addressed prior to publication of this manuscript.

1. In Figure 1, total STAT1 staining should also be shown for each cell type as well as phospho-STAT1, since this may increase because total STAT1 protein is also increased. The authors have done this in subsequent figures, but it should also be included here.
2. In Figure 2, many of the metabolic analyses have been carried out on cells that have been harvested and then restimulated with PMA/ionomycin. This is quite an unusual approach to restimulate cells in this manner prior to metabolic analysis and will likely alter the metabolic parameters measured. Justification of this approach should be provided, together with some description of the potential impact on cellular metabolic activity.
3. Figure 2 - The extracellular flux (Seahorse) data shown indicate that glycolysis, glycolytic reserve and glycolytic capacity are slightly increased in Th1 cells compared to "Act" cells, but this is really quite a small effect size, particularly when compared to the increase in these parameters in Act and Th1 cells vs. non-activated cells. This should be described in the text for clarity and to more faithfully describe the data shown - i.e. the phrase "ECAR was more strongly elevated in Th1 culture conditions" should be edited to acknowledge the small effect size. From these experiments, it would also be helpful to show OCR curves and summarised data which have been used to calculate the ATP rate assays (can be in the supplemental data).
4. Again in Figure 2 - In line 221-222 the authors state that "Th1 cell cultures depended on glycolysis, the reduction in puromycin incorporation by 2-deoxyglucose (2-DG) was stronger in Th1 cells" than Act cells, but this statement is not supported by the data provided in Figure 2g where calculation of glucose dependence shows this to be equal in both cell types - the description of the results should be edited. In addition, the summary statement at the end of the paragraph "these data support the hypothesis that anabolism of human Th1 cells is particularly fueled by glycolysis." should also be edited since the data as presented do not support this statement.
5. In Figure 2i the authors have assessed glucose uptake using the fluorescent glucose analogue 6-NBDG. This is structurally very similar to the molecule 2-NBDG which has previously been used for the same purpose. Importantly, several recent studies have reported that 2-NBDG is taken up in a non-specific manner by cells, which does not require glucose transporters and is not blocked in a competitive manner by glucose. It therefore does not report specific glucose uptake capacity. Whilst this has not been shown for 6-NBDG, it is likely to also be the case. I suggest these data should therefore be removed from the manuscript since this method is not reliable, or at the very least the reports highlighting the limitations of 2-NBDG be clearly discussed both at the point in the results where these data are shown and also within the discussion.
6. In Figure 2j the authors show cell culture supernatant concentrations of glucose and lactate as further evidence that Th1 cells are more glycolytic than Act cells - these values should be normalised to cell number, or at least shown alongside data on proliferative capacity of Act vs. Th1 cells, since the amount of glucose and lactate in the cell culture will reflect both the number and metabolic activity of the cells present.
7. The statement "these six lines of evidence show that human Th1 cells exhibit a high biometabolic activity which mainly depends on glycolysis, with excess glucose turnover, suggesting a relationship between human Th1 differentiation and glycolysis." Should be edited after changes to the data presented and described above to faithfully describe the alterations.
8. In Figure 4 there are currently no data provided to show the effects of the SHP2 inhibitors in isolation (i.e. in absence of 2-DG). It is important to include this to verify that they have capacity to increase STAT1 phosphorylation and IFN-gamma expression as expected at the concentrations employed, which would strengthen the conclusion that effects of 2-DG are independent of this.
9. Conversely, in Figure 5, effects of the two inhibitors are shown in isolation, but not together with 2-DG. Use of the agents in combination may better support the conclusion that glycolysis underpins this process (sensitive to 2-DG inhibition) if there is no additional effect of 2-DG on the readouts when the inhibitors are already present.
10. As noted above for Figure 2, in Figure 6, cells have also been stimulated with PMA/ionomycin prior to harvesting for STAT1 immunoprecipitation. Again this is a bit unusual and some justification for the approach would be helpful.
11. In Figure 7 and line 457 the authors describe an additional protocol with "late induction of differentiation". It's not very clear how this differs from the original protocol and what the justification for this is and it would be helpful to have some clearer detail on this.

12. In figures 3 and 7, example flow cytometry histograms for IFN-gamma indicate that the sample has been gated on IFN-gamma "bright" or high expressing cells (presence of abrupt cut-off to the left side of the histograms). Gating in this way is not described in the overall gating strategy provided and will impact the numerical data extracted and presented (IFN-gamma MFI of population). The gating strategy should be presented more clearly and justified with reassurance it does not impact the results shown.

Minor points:

For Figure 1, it would be helpful to have some brief details about the Th1 differentiation protocol provided in the figure legend. In the description of Figure 1, in line 158-159 the authors state "Moreover, Th1 cells produced more IFN- γ but not IL-4 or IL-17 (Figure 1c, d, e)". This implies that Th1 cells express less IL-4 and IL-17 than "Act" cells, but IL-4 and IL-17 expression are not shown for Th1 cells. The sentence should be rephrased to make more clear.

In line 194 there is a typo (va should be vs.)

In line 208 the phrase "The increase in glycolytic capacity and glycolytic reserve of Th1 cells indicated a higher rate of converting glucose into pyruvate or lactate" should be edited to remove the word pyruvate since this analysis only measures the amount of lactate produced (and hence extracellular acidification)

In line 217, use of the word "basal" for puromycin incorporation is confusing since it implies resting cells in this context, this should be edited for clarity to specify puromycin incorporation in absence of inhibitors.

In Figure 3, cells have been stimulated with PMA and ionomycin for 30 minutes prior to intracellular cytokine staining. This is quite unusual, since cells are typically stimulated for 4-6 hours with these reagents. It would be helpful to have some justification for this protocol.

Reviewer #3 (Comments to the Authors (Required)):

The study by Abir et al. underscores the critical role of glycolysis in maintaining Th1 lineage commitment by stabilizing STAT1 activity through OGT-mediated O-glycosylation. The authors conducted a comprehensive set of experiments using both pharmacological inhibitors and CRISPR-Cas9-mediated gene editing to interrogate the roles of glycolysis, O-glycosylation, and STAT1 in regulating the Th1 response. Notably, all experiments were performed in human PBMC-derived CD4⁺ T cells, which strengthens the physiological relevance of the findings. However, several limitations need to be addressed to further substantiate the conclusions:

1. The authors primarily relied on IFN- γ production to define the Th1 response. However, T-bet, a lineage-defining transcription factor for Th1 cells and a direct target of STAT1, is a crucial marker that should be evaluated in parallel. Including T-bet expression in all experimental conditions where IFN- γ levels were measured would provide more robust evidence for an effect on Th1 lineage commitment.

2. Although the observed ECAR differences between activated T cells and Th1 cells were statistically significant, the modest magnitude of change casts doubt on their biological relevance. Moreover, Figures 2g and 2i (glucose dependence and 6-NBDG uptake assays, respectively) did not reveal significant metabolic differences. These findings should be re-evaluated. A complete bioenergetic profile, including both ECAR and OCR, would provide a more comprehensive understanding of the metabolic state of Th1 cells.

3. Prior work (PMID: 23746840) demonstrated that glycolysis modulates IFN- γ production by sequestering GAPDH away from the 3' UTR of *Irfng*, preventing translational repression. It would be valuable to investigate whether a feed-forward loop exists between STAT1 and glycolysis. Using STAT1 knockout T cells could help determine whether STAT1 is involved in reinforcing glycolytic activity during Th1 differentiation.

4. The authors attribute the effects of 2-DG on IFN- γ production to glycolytic inhibition. However, 2-DG can also interfere with the HBP, which branches from glycolysis at fructose-6-phosphate and generates UDP-GlcNAc, a substrate for O-GlcNAcylation. To dissect the specific contribution of glycolysis versus the HBP, additional inhibitors such as iodoacetate (IAA; GAPDH inhibitor) and oxalate (pyruvate kinase inhibitor) should be tested.

5. While the data suggest that O-glycosylation of STAT1 is important for IFN- γ production and Th1 lineage stabilization, the precise mechanism remains unclear. Does glycosylation enhance STAT1 nuclear translocation, stability, or transcriptional activity? Do Th1 cells expressing wild-type versus glycosylation-deficient STAT1 differ in T-bet expression? Addressing these questions would elucidate the mechanistic basis of STAT1 glycosylation in Th1 differentiation and lineage fidelity.

Life Science Alliance Manuscript LSA-2025-03315-T – point to point reply

Reviewer #1 (Comments to the Authors (Required)):

1. Like the authors correctly indicated in "Introduction", page 4 line 111, "STAT1 is essential for Th1 cell differentiation and the production of IFN γ ". This has been well demonstrated both in mice and human T cells. Therefore, Figure 1 does not provide novel findings or novel information. As the authors described (page 6 line 160) "These data validate efficient human Th1 cell differentiation in our cultured cells", this figure should be presented as supplementary figure. The texts should be simplified accordingly.

This reviewer is right, we did not establish new findings here and shortened this paragraph accordingly. However, we kept this figure in the main file as it introduces our Crispr/Cas9 approach, which we pick up again in the very last figure.

2. The authors concluded that "2-DG-induced STAT1 dephosphorylation partially depends on SHP2" (page 12 line 320), 2-DG may intrinsically activate SHP2. They showed that "SHP2 inhibition elevated the phosphorylation "(page 13 line 329). Since the inhibitor may have other effects, they should have shown by western blot the SHP2 activation, e.g. by detecting the pSHP2.

True, inhibitors may be unspecific or have off-target effects. It is important to acknowledge these possibilities. Therefore, we used two SHP2 inhibitors, TNO155 and RMC, with very similar results. Hence, we consider it unlikely that the two SHP2 inhibitors bypass SHP2 activation. Otherwise, there would be no increase in STAT1 pTyr701 phosphorylation.

3. In Figure 4b, c, d, statistical analysis of the two SHP2 inhibition versus 2-DG should be included.

Statistical analysis of the two SHP2 inhibition versus 2-DG has been included in Figure 4 d.

4. As one of the most important conclusions, the authors claimed that OGT inhibition significantly decreased IFN γ producing cells in Figure 5d-f. What concentrations of the two inhibitors BADGP and OSMI1 were used? The authors should present data showing the reduced IFN γ producing cells are not due to the increased cell death? Furthermore, the authors should provide data showing the dose dependent effect of these two inhibitors.

We added the concentrations of BADGP and OSMI1 in the Materials & Methods section. We relied on previously established concentrations causing no cell death (Jitschin et al, 2019; PMID: 30679801; Liu et al., 2017; PMID: 28951553). Along this line, we included live/dead stainings using Zombie Aqua (please see updated Supplemental Figure S1). We did not observe increased cell death to a rate that it would affect our conclusions. Because we gated on live cells, reductions in any parameter (IFN γ , phosphorylation, glycosylation) cannot be due to cell intrinsic cell death.

5. In Figure 7, how were the p values calculated? The information on statistical methods used in each of the comparisons is missing. Please add it. Please provide the raw data of each IFN γ MFI and percentage in Fig7b---i on a supplementary table.

We updated the statistical information in the figure legend and raw data table is also been included in the supplementary (Supplementary Table 2).

Reviewer #2 (Comments to the Authors (Required)):

1. In Figure 1, total STAT1 staining should also be shown for each cell type as well as phospho-STAT1, since this may increase because total STAT1 protein is also increased. The authors have done this in subsequent figures, but it should also be included here.

We agree with this comment, the abundance of total protein can influence the relative degree of its phosphorylation. We have included the data (Figure 1j).

2. In Figure 2, many of the metabolic analyses have been carried out on cells that have been harvested and then restimulated with PMA/ionomycin. This is quite an unusual approach to restimulate cells in this manner prior to metabolic analysis and will likely alter the metabolic parameters measured. Justification of this approach should be provided, together with some description of the potential impact on cellular metabolic activity.

That's a valid point, of course. Ionomycin and PMA stimulation reactivates T cells *ex vivo* and enables the detection of intracellular cytokines by enhancing their transcription and translation. PMA can in fact influence T cell metabolism, specifically glycolysis (Menk et al., 2018; PMID: 29425506), albeit at a lesser extent than TCR/CD28 activation and bypassing PDHK1. There are scarce studies about the effect of ionomycin on T cell metabolism, or they escaped our attention. Generally, ionomycin amplifies calcium-dependent activation pathways. In neutrophils, ionomycin can re-activate mitochondrial metabolism and fuel the import of glycolytic intermediates into the TCA cycle (Lika et al., 2025; PMID: 40356902). Overall, PMA and ionomycin together might influence glycolysis in T cells with ongoing or increased OxPhos. We have added the following sentence to the respective paragraph: **However, to be able to contextualize these and the following experiments we used the same conditions as for the IFN γ staining, that is, 4h of PMA/Ionomycin stimulation.**

In our hands, previous T cell activation elicits a much stronger metabolic response than PMA/Ionomycin alone (please compare naïve with activated T cells in Figure 2a). Hence, we believe that T cell activation dominates the metabolic response over PMA/Ionomycin. On the other hand, PMA/Ionomycin are required to enable intracellular cytokine staining by several mechanisms. In attempts to delineate metabolic changes on the single cell level we defined IFN γ production as the gold standard for human Th1 cell identity (Figure 2f-j). To this end, stimulation of the cells with PMA/Ionomycin was necessary.

We have qualified the respective paragraphs.

3. Figure 2 - The extracellular flux (Seahorse) data shown indicate that glycolysis, glycolytic reserve and glycolytic capacity are slightly increased in Th1 cells compared to "Act" cells, but this is really quite a small effect size, particularly when compared to the increase in these parameters in Act and Th1 cells vs. non-activated cells. This should be described in the text for clarity and to more faithfully describe the data shown - i.e. the phrase "ECAR was more strongly elevated in Th1 culture conditions"

should be edited to acknowledge the small effect size. From these experiments, it would also be helpful to show OCR curves and summarised data which have been used to calculate the ATP rate assays (can be in the supplemental data).

Glycolysis, glycolytic reserve and glycolytic capacity are in fact only slightly increased in Th1 cells and we put that more into perspective: ... ECAR was slightly more elevated in Th1 culture conditions (Figure 2a, b). This mildly elevated glycolytic capacity and glycolytic reserve of Th1 cells indicated a higher rate of converting glucose into lactate (Figure 2c, d). ATP rate assays confirmed that Act.T, but more so Th1 cells...

We have added the OCR curves used to do the ATP rate analysis to the Supplement. The ATP rate calculation has been performed with these and the ECAR data in the "WAVE" software routine application.

4. Again in Figure 2 - In line 221-222 the authors state that "Th1 cell cultures depended on glycolysis, the reduction in puromycin incorporation by 2-deoxyglucose (2-DG) was stronger in Th1 cells" than Act cells, but this statement is not supported by the data provided in Figure 2g where calculation of glucose dependence shows this to be equal in both cell types - the description of the results should be edited. In addition, the summary statement at the end of the paragraph "these data support the hypothesis that anabolism of human Th1 cells is particularly fueled by glycolysis." should also be edited since the data as presented do not support this statement.

Overall, the increment of the Δ Puro MFI was larger upon 2DG treatment in Th1 than in activated T cells because they appear to be more active, be it in separate or same cultures (Fig. 2fi and 2fii). However, it is absolutely correct that the calculated glucose dependence of protein translation does not differ between Act and Th1 cells. We edited the description of the results.

5. In Figure 2i the authors have assessed glucose uptake using the fluorescent glucose analogue 6-NBDG. This is structurally very similar to the molecule 2-NBDG which has previously been used for the same purpose. Importantly, several recent studies have reported that 2-NBDG is taken up in a non-specific manner by cells, which does not require glucose transporters and is not blocked in a competitive manner by glucose. It therefore does not report specific glucose uptake capacity. Whilst this has not been shown for 6-NBDG, it is likely to also be the case. I suggest these data should therefore be removed from the manuscript since this method is not reliable, or at the very least the reports highlighting the limitations of 2-NBDG be clearly discussed both at the point in the results where these data are shown and also within the discussion.

We acknowledge the concern regarding potential receptor-independent uptake of 6-NBDG and 2-NBDG, suggesting NBDG uptake may not represent glucose uptake (Hamilton et al., 2021; PMID: 34224807). In contrast, other studies have shown that 6-NBDG exhibits a markedly high binding affinity to GLUT1 - approximately 300-fold higher than glucose (Barros et al., 2009; PMID: 19393014), suggesting that it does function through transporter-mediated mechanisms. Along this line, we have recently shown that 2-NBDG uptake does depend on the glucose transport GLUT1, at least in primary B cells (Bierling et al., 2024; PMID: 38340319). In alignment with the editor's guidance, we have chosen to retain the 6-NBDG data in Figure 2 **while discussing its limitations in the revised manuscript**: To substantiate these findings we performed a **surrogate** glucose uptake assay. After PMA/ionomycin restimulation, the cells were incubated in glucose-free medium with 6-NBDG (6-(N-(7-Nitrobenz-2-oxa-1,3-diazol-4-yl)amino)-6-Deoxyglucose), a fluorescent glucose analogon, for 30 min. Th1 cells showed only a slightly higher median fluorescence intensity (MFI) for 6-NBDG (Figure 2i). However, glucose and lactate abundance in the culture medium on day 5

of the culture period revealed that Th1 cultures contained much less glucose and more lactate, evidencing a strong glycolytic activity (Figure 2j). These data suggested that the small differences in surrogate glucose uptake are disproportional in relation to the metabolic consequences, which might be explained by receptor independent uptake of glucose (Hamilton et al., 2021). To reconcile these findings we monitored glucose driven NADH generation in real time by flow cytometry after adding glucose to starved cells (Abir et al., 2024) (Figure 2k), demonstrating that acute glycolytic flux is higher in Th1 than Act.T cells. Together, human Th1 cells exhibit a high biometabolic activity biased towards glycolysis, with excess glucose turnover, suggesting a relationship between human Th1 differentiation and glycolysis....

We also edited the abstract accordingly.

We cannot completely exclude that T cells take up 6-NBDG in a receptor independent manner. Yet, even if they did so, our results would not change.

6. In Figure 2j the authors show cell culture supernatant concentrations of glucose and lactate as further evidence that Th1 cells are more glycolytic than Act cells - these values should be normalised to cell number, or at least shown alongside data on proliferative capacity of Act vs. Th1 cells, since the amount of glucose and lactate in the cell culture will reflect both the number and metabolic activity of the cells present.

We agree with this comment. We have normalized the data to the actual cell numbers.

7. The statement "these six lines of evidence show that human Th1 cells exhibit a high biometabolic activity which mainly depends on glycolysis, with excess glucose turnover, suggesting a relationship between human Th1 differentiation and glycolysis." Should be edited after changes to the data presented and described above to faithfully describe the alterations.

We agree and have edited this statement.

8. In Figure 4 there are currently no data provided to show the effects of the SHP2 inhibitors in isolation (i.e. in absence of 2-DG). It is important to include this to verify that they have capacity to increase STAT1 phosphorylation and IFN-gamma expression as expected at the concentrations employed, which would strengthen the conclusion that effects of 2-DG are independent of this.

We understand this argument. These data are included now and we cannot observe elevated STAT1 phosphorylation or IFN γ production after SHP-2 inhibition only. This is in fact not expected. Although unlikely, there might be no basal SHP-2 activity under the employed conditions. Perhaps SHP-2 activity is suppressed by PMA/Ionomycin through oxidation by ROS, or S-nitrosylation of the catalytic cysteine; Perez-Fernandez et al., 2019, PMID: 30764849; Barrett et al., 2005, PMID: 15684422), but 2-DG treatment might overcome this putative scenario. We added a comment in the manuscript.

9. Conversely, in Figure 5, effects of the two inhibitors are shown in isolation, but not together with 2-DG. Use of the agents in combination may better support the conclusion that glycolysis underpins this process (sensitive to 2-DG inhibition) if there is no additional effect of 2-DG on the readouts when the inhibitors are already present.

Here, we have a different opinion. Because 2-DG, BADGP and OSMI1 obviously block the same pathway, albeit at different hierarchic levels, with BADGP and OSMI1 being more specific and efficient, simultaneous treatment would likely not provide additional information.

10. As noted above for Figure 2, in Figure 6, cells have also been stimulated with PMA/ionomycin prior to harvesting for STAT1 immunoprecipitation. Again this is a bit unusual and some justification for the approach would be helpful.

We understand the reviewer's irritation. We would like to argue that we tried to keep the experimental conditions very similar throughout the paper (please see previous comments) as our conservative definition of human Th1 cells is the detection of intracellular IFN γ .

11. In Figure 7 and line 457 the authors describe an additional protocol with "late induction of differentiation". It's not very clear how this differs from the original protocol and what the justification for this is and it would be helpful to have some clearer detail on this.

The STAT1 mutation was introduced after 48 hours of culture. In the conventional Th1 culture system (early differentiation), IL12 is already present during the initiation of the culture. In the "late differentiation" protocol, activated T cells are treated with IL12 only after the electroporation was performed, that is, after 48h. The goal to include the late differentiation protocol was to see how the cells perform with a mutated STAT1 in terms of their capability to produce IFN γ and to sustain the Th1 lineage compared to the ones that have been treated with IL12 for 48h before electroporation.

12. In figures 3 and 7, example flow cytometry histograms for IFN-gamma indicate that the sample has been gated on IFN-gamma "bright" or high expressing cells (presence of abrupt cut-off to the left side of the histograms). Gating in this way is not described in the overall gating strategy provided and will impact the numerical data extracted and presented (IFN-gamma MFI of population). The gating strategy should be presented more clearly and justified with reassurance it does not impact the results shown.

To avoid gating-related confusion and to mitigate the concern about the 'abrupt cut-off' of the population, we included the representative pre-gate that was used for Figure 2-5 in the Supplementary Figure 2. We also simplified the gating in Figure 7b and Supplementary Figure 4a.

Minor points:

For Figure 1, it would be helpful to have some brief details about the Th1 differentiation protocol provided in the figure legend

We did include some brief details.

In the description of Figure 1, in line 158-159 the authors state "Moreover, Th1 cells produced more IFN-y but not IL-4 or IL-17 (Figure 1c, d, e)". This implies that Th1 cells express less IL-4 and IL-17 than "Act" cells, but IL-4 and IL-17 expression are not shown for Th1cells. The sentence should be rephrased to make more clear.

We specified this sentence: ... fold more IFN γ ⁺ cells than Act.T cells while IL-4 or IL-17 were absent and T-bet was more abundant **in the IFN γ ⁺ cells.**

In line 194 there is a typo (va should be vs.)

Thank you. We corrected this typo.

In line 208 the phrase "The increase in glycolytic capacity and glycolytic reserve of Th1 cells indicated a higher rate of converting glucose into pyruvate or lactate" should be edited to remove the word pyruvate since this analysis only measures the amount of lactate produced (and hence extracellular acidification)

Thank you. We removed the word pyruvate.

In line 217, use of the word "basal" for puromycin incorporation is confusing since it implies resting cells in this context, this should be edited for clarity to specify puromycin incorporation in absence of inhibitors.

Thank you. We have clarified this sentence.

In Figure 3, cells have been stimulated with PMA and ionomycin for 30 minutes prior to intracellular cytokine staining. This is quite unusual, since cells are typically stimulated for 4-6 hours with these reagents. It would be helpful to have some justification for this protocol.

This notion is correct. Our description was unclear. We stimulated the cells as usual and just added the 2-DG and Oligomycin for 30 min. We have corrected this issue in the figure legend.

Reviewer #3 (Comments to the Authors (Required)):

1. The authors primarily relied on IFN- γ production to define the Th1 response. However, T-bet, a lineage-defining transcription factor for Th1 cells and a direct target of STAT1, is a crucial marker that should be evaluated in parallel. Including T-bet expression in all experimental conditions where IFN- γ levels were measured would provide more robust evidence for an effect on Th1 lineage commitment.

This argument is well taken. We did evaluate T-bet expression (Figure 1f, g), showing increased abundance in Th1 cells. This supports the validity of our differentiation protocol. We also included it in our analyses of STAT1mut vs. STA1wt cells but did not add it to the paper (please see comment 6). One might argue whether IFN γ or T-bet is the best marker for human Th1 cells. It has been shown, for instance, that T-bet and IFN γ expression in human T cells do not necessarily correlate (Ylikoski et al., 2005; PMID: 16220539). Our approach of relying predominantly on IFN γ expression as Th1-defining marker is in our view on the more conservative side.

2. Although the observed ECAR differences between activated T cells and Th1 cells were statistically significant, the modest magnitude of change casts doubt on their biological relevance. Moreover, Figures 2g and 2i (glucose dependence and 6-NBDG uptake assays, respectively) did not reveal significant metabolic differences. These findings should be re-evaluated. A complete bioenergetic profile, including both ECAR and OCR, would provide a more comprehensive understanding of the metabolic state of Th1 cells.

Indeed, Th1 cells show only slightly more ECAR than Act T cells. It is also correct that activated (non-polarized) and Th1 cells both show a strong dependence on glucose consumption for protein synthesis (Puromycin incorporation). Overall, activated T cells and Th1 cells do not appear to differ qualitatively regarding general parameters of metabolism. However, Th1 cells produce more lactate per cell, showing that the glucose that was taken up branches towards different pathways in activated T cells vs. Th1 cells. We qualified the paragraph describing these results to be more precise, integrating here also comments from reviewer 1. To this end, we also provide the OCR in the Supplement. Yet, we believe that glycolysis does influence in Th1 cells the glycosylation of STAT1, and thereby, its phosphorylation. Overall, we propose that this mechanism ensures Th1 identity.

3. Prior work (PMID: 23746840) demonstrated that glycolysis modulates IFN- γ production by sequestering GAPDH away from the 3' UTR of *Ifng*, preventing translational repression. It would be valuable to investigate whether a feed-forward loop exists between STAT1 and glycolysis. Using STAT1 knockout T cells could help determine whether STAT1 is involved in reinforcing glycolytic activity during Th1 differentiation.

This is an interesting proposal aligning with our general perception of how STAT1 functions. Indeed, Pitroda and colleagues have shown that STAT1 transactivates many genes involved in glycolysis and gluconeogenesis (Pitroda et al., 2009; PMID: 19891767). Thus, there might be multiple feedback loops. Although very appealing, currently we do not see the chance to generate sufficient numbers of pure primary STAT1 KO T cells to do meaningful analyses. Yet we plan to screen patients from our Immunology and Rheumatology Department for activating and inactivating STAT1 mutations to follow up on this.

4. The authors attribute the effects of 2-DG on IFN- γ production to glycolytic inhibition. However, 2-DG can also interfere with the HBP, which branches from glycolysis at fructose-6-phosphate and generates UDP-GlcNAc, a substrate for O-GlcNAcylation. To dissect the specific contribution of glycolysis versus the HBP, additional inhibitors such as iodoacetate (IAA; GAPDH inhibitor) and oxalate (pyruvate kinase inhibitor) should be tested.

We thank the reviewer for this proposal and have performed additional experiments accordingly. Both IAA and oxalate reduced Th1 cell differentiation as well. We have included a new supplemental figure including a glycolysis flow chart (S3) that depicts the hierarchical level of inhibitor action. IAA and oxalate act both downstream of HBP branching. Their effects appear to be smaller than that of 2-DG. It seems that the more downstream the inhibitor acts, the less effective it is. Most reactions in the glycolysis pathway are equilibrium reactions. However, Phosphofructokinase, which is downstream of HBP branching, prefers the reaction from Fructose-6-phosphate to Fructose-1,6-phosphate. Therefore, more downstream inhibitors may be less effective in inhibiting glycolysis upstream of Phosphofructokinase. Overall, these data are compatible with our interpretation that 2-DG likely exerts at least some of its inhibitory effect on Th1 cell differentiation by acting on the HBP/OGT pathway. **We added paragraphs to the results and discussion section of the main manuscript file**

6. While the data suggest that O-glycosylation of STAT1 is important for IFN- γ production and Th1 lineage stabilization, the precise mechanism remains unclear. Does glycosylation enhance STAT1 nuclear translocation, stability, or transcriptional activity? Do Th1 cells expressing wild-type versus glycosylation-deficient STAT1 differ in T-bet expression? Addressing these questions would elucidate the mechanistic basis of STAT1 glycosylation in Th1 differentiation and lineage fidelity.

These are important questions that we have started to address. We found that Th1 cells expressing mutated STAT1 contain less T-Bet than those where the *stat1* gene was replaced by an unmutated *stat1* gene (Figure for review only).

This result underpins our interpretation and implicates that ultimately, transcriptional activity of the mutated STAT1 is lowered. As pointed out by this reviewer, this might be due to inefficient nuclear translocation, altered protein stability, DNA binding, transactivation potential. Certainly, scRNA expression analysis, RNASeq of GFP+/- sorted cells or cut & run assays would answer some of the above mentioned questions but we believe that this is for now beyond the scope of this manuscript.

October 13, 2025

RE: Life Science Alliance Manuscript #LSA-2025-03315-TR

Prof. Dirk Mielenz
Friedrich-Alexander-Universität Erlangen-Nürnberg
Division of Molecular Immunology
Glueckstr. 6
Erlangen, Bavaria 91054
Germany

Dear Dr. Mielenz,

Thank you for submitting your revised manuscript entitled "Glycolytic flux sustains human Th1 identity and effector function via STAT1 glycosylation". Your revised manuscript was evaluated by one of the original reviewers whose comments are appended below. As you will read, the reviewer notes that the revised manuscript has addressed the previous concerns. However, there are two open points from the original reviewers who did not review the revised manuscript. We encourage you to address these:

- In the discussion of results, we suggest that you include your rationale of using 2 distinct inhibitors while commenting on the lack of evidence on pSHP2 (as raised by Reviewer 1, in point 2).
- In the description of the experimental approach for results described in Figure 7 (lines 316-319), we suggest that you to include terminology consistent with the corresponding description in the methods (line 646 onwards; i.e. early and late differentiation stages; raised by Reviewer 2, point 11).

We would be happy to publish your paper in Life Science Alliance pending final revisions necessary to meet our formatting guidelines. Along with points mentioned below, please tend to the following:

- We encourage you to name the paired t-test in the legends consistent with the methods, and encourage you to include a rationale for your choice of the different statistical tests.
- Thank you for providing the details of antibodies and reagents used in this work. We encourage you to refer to the table as follows "Materials and Methods (Detailed information is in the supplemental section)".
- Please upload the graphical abstract as a single file with the file designation "Graphical Abstract" and remove it from the manuscript file
- Please upload a clean manuscript file without the highlighted text.
- Please consult our guidelines on the "Data Availability" section at <https://www.life-science-alliance.org/manuscript-prep#format>
- Please be sure that all authors are mentioned in the Author's Contribution section of the manuscript file
- The contributions selected for Gerhard Krönke do not qualify them for authorship. Please either update the contributions in our system and the Author Contributions section of the manuscript, or let us know if the author needs to be removed (and added eventually to the acknowledgment section)
- Please use the [10 author names, et al.] format in your references (i.e., limit the author names to the first 10)
- We encourage you to revise the figure legends for figures 3, S2, and S3 such that the figure panels are introduced in alphabetical order.
- Tables should be numbered consecutively with Arabic numerals (1, 2, 3, 4); They can be included at the bottom of the main manuscript file or be sent as separate files.
- Please add callouts for Figures 3G; 4D and E; S1A-C; S2A, B; S3C, D; S4A-H and tables 2-5 to your main manuscript text
- Please upload all figure files as individual ones, including the supplementary figure files; all figure legends should only appear in the main manuscript file
- Please add the X and Bluesky handles of your host institute/organization, as well as your own and/or one of the authors in our system
- Please be sure that the authorship listing and order is correct

LSA now encourages authors to provide a 30-60 second video where the study is briefly explained. We will use these videos on social media to promote the published paper and the presenting author (for examples, see <https://docs.google.com/document/d/1-UWCfbE4pGcDdcgzcmiuJl2XMBJnxKYeqRvLLrLS08s/edit?usp=sharing>). Corresponding or first-authors are welcome to submit the video. Please submit only one video per manuscript. The video can be emailed to contact@life-science-alliance.org

A. FINAL FILES:

B. MANUSCRIPT ORGANIZATION AND FORMATTING:

Thank you for your attention to these final processing requirements. Please revise and format the manuscript and upload materials as soon as you are able.

Sincerely,

Sarita Hebbar, PhD
Scientific Editor
Life Science Alliance
<http://www.lsjournal.org>

Reviewer #3 (Comments to the Authors (Required)):

The authors have adequately addressed all the concerns raised. The manuscript is suitable for acceptance and publication.

October 21, 2025

RE: Life Science Alliance Manuscript #LSA-2025-03315-TRR

Prof. Dirk Mielenz
Friedrich-Alexander-Universität Erlangen-Nürnberg
Department of Translational Immunology
Glueckstr. 6
Erlangen, Bavaria 91054
Germany

Dear Dr. Mielenz,

Thank you for submitting your Research Article entitled "Glycolytic flux sustains human Th1 identity and effector function via STAT1 glycosylation". It is a pleasure to let you know that your manuscript is now accepted for publication in Life Science Alliance. Congratulations on this interesting work.

Your manuscript will now progress through copyediting and proofing. When dealing with the proofs, I remind you to kindly resolve the following open issue from last time: to revise the figure legends for figures S2 and S3 such that the figure panels are introduced in alphabetical order.

It is journal policy that authors provide original data upon request.

DISTRIBUTION OF MATERIALS:

Again, congratulations on a very nice paper. I hope you found the review process to be constructive and are pleased with how the manuscript was handled editorially. We look forward to future exciting submissions from your lab.

Sincerely,

Sarita Hebbar, PhD
Scientific Editor
Life Science Alliance
<http://www.lsajournal.org>